

# A direct comparison of single grain and multi-grain aliquot luminescence dating of feldspars from colluvial deposits in KwaZulu-Natal, South Africa

Svenja Riedesel[1,2], Guillaume Guérin[3], Kristina J. Thomsen[1], Mariana Sontag-González[4],
Matthias Blessing[5,6], Greg A. Botha[7], Max Hellers[8], Gunther Möller[5], Andreas Peffeköver[2], Christian Sommer[9,10], Anja Zander[2], Manuel Will[5,11]

[1]Radiation Physics Division, Department of Physics, Technical University of Denmark, Roskilde/Lyngby, Denmark
[2]Institute of Geography, University of Cologne, Cologne, Germany
[3]Géosciences Rennes, UMR 6118, CNRS, Bâtiment 15, Campus Beaulieu, Université de Rennes 1, 35042 Rennes, France
[4]Institute of Geography, Justus-Liebig-Universität Giessen, D-35390 Gießen, Germany
[5]Department of Early Prehistory and Quaternary Ecology, University of Tübingen, 72070 Tübingen, Germany
[6]Deep History Lab, Department of Anthropology, University of Connecticut, Storrs, CT, USA
[7]Geological Sciences, School of Agricultural, Earth and Environmental Sciences, University of KwaZulu-Natal, Westville, South Africa
[8]Institute of Geology and Mineralogy, University of Cologne, Cologne, Germany
[9]Institute of Geography, Department of Geosciences, University of Tübingen
[10]The Role of Culture in Early Expansions of Humans, Heidelberg Academy of Sciences and Humanities, Tübingen, Germany
[11]Palaeo-Research Institute, University of Johannesburg, P.O. Box 524, Auckland Park ZA-2006, Johannesburg, South Africa

*Correspondence to*: Svenja Riedesel (riedeselsvenja@gmail.com)

**Abstract.**

The erosional landscape of the Jojosi Dongas in KwaZulu-Natal, South Africa, expose accretionary slope deposits that preserve important geological and archaeological information. This landscape has been occupied by modern humans during the stone age for many thousands of years as evidenced by the presence of numerous stone artefacts interbedded within at least three phases of gully cut-and-fill deposits. A contextualisation of the artefacts and their role for human evolution in southern Africa, as well as developing an understanding of the environmental conditions that shaped this inhabited landscape, is only possible by establishing a robust chronological framework.

Here we use luminescence dating of feldspars to constrain the geochronological framework for the sequence of accretionary hillslope deposition at Jojosi at three sampling locations. Initial suspicion of poor bleaching led us to measure single grains of feldspar, which revealed low luminescence sensitivity





of the individual grains and a variable proportion of grains in saturation. Summing the luminescence

signal of individual grains and creating synthetic aliquots enabled us to study the effect of signal averaging on the luminescence sensitivity, signal saturation, and dose distributions. We then compare the results from individual grain measurements and synthetic aliquots to true multi-grain aliquots. To allow for a quantification of the results, we apply four different dose models to the distributions, including the Central Age Model, the Average Dose Model, BayLum, and a standardised growth curve (SGC) approach using

an averaged $L_n/T_n$ value interpolated onto the SGC. Doses calculated for the different samples range from ~80 Gy to ~800 Gy and contain up to 67 % saturated grains. We evaluate the performance of the different dose models over this large dose range, with samples close to the saturation level of feldspar luminescence.

On average we find good agreement between the results obtained using the different dose models, but

observe that samples with a large number of saturated grains impact the consistency of the result. Overall, all dose models and data sets give consistent results below a saturated grain threshold of ~15 %, corresponding to a dose of ~120 Gy in this study.

Finally, we favoured BayLum for age calculations of the single grain and multi-grain aliquot data sets, representing the opportunity to refine the chronology by including stratigraphic information in the age

calculations. We were able to establish a chronology for the three sampled sections within the Jojosi Dongas constraining erosional and depositional processes from ~100 ka to ~700 ka, and human occupation of the area in the early MIS 5 and late MIS 6.

## 1 Introduction

Optically stimulated luminescence (OSL) dating is a widely used geochronological technique to constrain

the last exposure to sunlight of sediment prior to its deposition and burial, and it is thus able to provide chronologies in archaeological and geological contexts (Huntley et al., 1985; Murray et al., 2021). Here we applied luminescence dating to stratified hillslope sediments exposed by "donga" incision which is an isiZulu name for gully erosion features in South Africa. Due to their wide-spread occurrence in central KwaZulu-Natal and deep incision of hillslope regolith mantle, dongas are of geological, and



archaeological interest within southern Africa (Poesen et al., 2003; Mararakanye and Le Roux, 2012; Olivier et al., 2023). We focus on the Jojosi Dongas located north of Nqutu in northern-central KwaZulu-Natal, Soth Africa, which hold important geological and archaeological archives (Botha et al., 1994; Botha, 1996; Will et al., 2024; Möller et al., in prep). The erosional landscape has seen a complex interplay of phases dominated by erosion, gully incision and sediment accretion evidenced by an accretionary

succession of gully cut-and-fill deposits that likely reflects variation in landscape stability during the Pleistocene. The Jojosi catchment is dominated by dolerite-derived colluvium and interbedded palaeosols that have precluded direct correlation with the sequence of colluvial sedimentary units and palaeosols described from the surrounding region (Botha et al., 1994; Botha, 1996) The temporal range of colluvial slope processes has also been constrained using luminescence dating by Li (1992), Li and Wintle (1991,

1992), Clarke et al. (2003). Temme et al. (2008), Lyons et al. (2013), and Colarossi et al. (2020).

The presence of numerous stone artefacts on the gully sidewalls and rarer archaeological material interbedded within colluvial sediments and palaeosols exposed by the gully sidewalls demonstrates the occupation of this landscape by humans during the Stone Age over for many thousands of years. The stratified stone artefacts derive from the Middle Stone Age, the period during which *Homo sapiens*

evolved on the African continent. Contextualisation of these stone tools in relation to long-term hillslope processes and evaluating their role for human evolution in southern Africa is only possible within a robust chronometric framework. A detailed introduction to the geology and archaeology of the Jojosi Dongas has been published elsewhere (see Will et al., 2024).

OSL dating is the ideal technique to constrain the depositional age of the hillslope deposits at Jojosi, as it

directly dates the last exposure to sunlight of sheetwash transported sediment prior to burial, along with stone artefacts created on the hillslope surface. Furthermore, by comparing different luminescence signals, as well as dose distributions for different aliquot sizes, information can be gained on the nature of the sediment studied (e.g. Duller, 2008). Quartz and feldspar act as natural luminescence dosimeters and are ubiquitous in most environments, with quartz being often preferred over feldspar. Firstly, the

quartz OSL signal resets faster during exposure to sunlight (e.g. Godfrey-Smith et al., 1988), and secondly, feldspar exhibits an anomalous loss of signal over time (fading, Wintle, 1973; Spooner, 1994),





which can lead to age underestimation if uncorrected. However, methods exist which either allow correction for fading (Huntley and Lamothe, 2001; Huntley, 2006; Kars et al., 2008), or which circumvent fading (e.g. Thomsen et al., 2008; Li and Li, 2011). Despite these advantages, quartz OSL has limitations,

i.e. earlier signal saturation limiting the age range (e.g. Wintle and Murray, 2006; Buylaert et al., 2012), or insufficient luminescence sensitivity (e.g. Rhodes, 2007; Duller, 2008; Mineli et al., 2021). Feldspar, on the contrary, has been shown to mostly exhibit bright luminescence signals (e.g. Baril and Huntley, 2003; Lamothe et al., 2012), allowing for its use in various geological contexts (e.g. Gliganic et al., 2017; Sawakuchi et al., 2018), as well as for dating even very young (tens of years to a few hundreds of years)

samples (e.g. Reimann et al., 2012; Riedesel et al., 2018; Buckland et al., 2019). Combining both quartz and feldspar luminescence enables cross validating the luminescence dating results and providing additional insights into signal resetting (e.g. Murray et al., 2012; Colarossi et al., 2015, 2020). However, not all geological settings allow such an inter-method comparison, e.g., when the quartz OSL signal is saturated or when it shows insufficient luminescence sensitivity.

Here we explore a situation where (i) due to source rock mineralogy (dolerite), too little quartz is present, and (ii) where the lack of any other datable material, such as volcanic ashes, charcoal, bone or teeth, prevents us from establishing a chronology independent of luminescence ages. We thus employ post-infrared infrared stimulated luminescence (post-IR IRSL) dating of feldspars to establish a chronology of the succession of colluvial deposits exposed at Jojosi. We use means provided by the feldspar post-IR

IRSL measurements, to enable an internal validation: (1) The post-$IR_{50}$ $IRSL_{225}$ protocol (Thomsen et al., 2008; Buylaert et al., 2009) facilitates recording of two luminescence signals, i.e. the lower temperature $IRSL_{50}$ signal, and the elevated temperature post-IR $IRSL_{225}$ signal. These two signals recorded using a single protocol enable us to explore two luminescence signals with different properties: The $IRSL_{50}$ signal is reset more rapidly by sunlight (e.g., Buylaert et al., 2012; Colarossi et al., 2015) but has also been

shown to exhibit larger fading rates, compared to the post-IR $IRSL_{225}$ signal (e.g. Thomsen et al., 2008). We use this to our advantage to check for signs of incomplete resetting of the post-IR $IRSL_{225}$ signal (cf. Buylaert et al., 2013). (2) The sediments to be dated at the Jojosi donga originate from weathered dolerite exposed proximally to the sediment deposits that have been transported over short distances prior to burial





and possibility mobilised repeatedly. Consequently, there is the suspicion that the inherited luminescence
signal in some of the sediment grains at Jojosi, might not have been fully reset prior to deposition and
burial. As an additional check for signs of incomplete bleaching, we decided to measure single grains of
feldspar which might also inform on potential influences from beta dose rate heterogeneity (e.g. Nathan
et al., 2003; Jankowski and Jacobs, 2018; Smedley et al., 2020). However, due to the low sensitivity of
the feldspar luminescence signal of our samples, it was necessary to measure many single grain discs to
obtain a sufficiently large dataset for our single grain analysis. Whilst this is time consuming, it also
enables us to directly compare equivalent dose distributions based on single grains, and those obtained
from synthetic aliquots created from these single grains using the Analyst software (c.f. Duller, 2015). (3)
Furthermore, we measured small multi-grain aliquots of all our samples to following a more conventional
approach. (4) Finally, we compare these different data sets and evaluate our results by applying different
dose models, including the Central Age Model (CAM, Galbraith et al., 1999), the Average Dose Model
(ADM, Guérin et al., 2017), BayLum (Philippe et al., 2019) and a standardised growth curve (SGC)
approach utilising the $L_nT_n$ method (Li et al., 2017, 2020), combined with the CAM. Comparing results
obtained from applying these different dose models will inform us on (i) the impact of scatter in the dose
distributions, and (ii) the effect of saturated grains on burial dose and age calculations.

## 2 Materials and methods

### 2.1 Sample collection and preparation

The samples were collected during field campaigns in 2022 and 2023 in the Jojosi Donga system (Fig. 1)
near Nqutu, KwaZulu-Natal, South Africa, within the framework of a combined geographical and
archaeological study investigating Middle Stone Age open-air sites and the dynamics between changing
landscapes and human activities (cf. Will et al., 2024). The samples were either collected in opaque
luminescence sampling tubes by hammering the tubes into sediment sections exposed donga sidewalls or
by carving out blocks from the exposed sections. Respective samples for dosimetry were taken within the
sediment surrounding the sampling tubes, accounting for visible variations in the sediment (i.e. grain size)
which might influence the gamma dose rate delivered to the samples. Nine samples from three different



sections were selected for this study that include sites of geographical ("Jojosi Triple Junction") and archaeological ("Jojosi 1", "Jojosi 5") interest (Fig. 1).

The samples were prepared under subdued red-light conditions in the Cologne Luminescence Laboratory (CLL, University of Cologne). Hydrochloric acid (HCl, 10 %) and hydrogen peroxide ($H_2O_2$, 10 %) were used to remove carbonates and organic material, respectively. Sodium oxalate ($Na_2C_2O_4$, 0.01 N) was

used to disperse the sediment particles. After chemical treatment the samples were sieved to obtain the 200-250 µm grain size fraction. From this fraction feldspar-rich extracts were separated using a sodium polytungstate solution at a density of 2.58 g cm$^{-3}$.

For pre-tests, performed to determine the appropriate measurement protocol, multi-grain aliquots (4 mm in diameter) of the isolated feldspar fraction were mounted on stainless steel discs using silicone oil. For

single grain equivalent dose determination, individual feldspar grains were brushed into 300 µm holes of single grain discs. To obtain multi-grain equivalent dose distributions, feldspars were mounted as 1 mm multi-grain aliquots (1 mm diameter, resulting in approximately <30 grains on a single disc; Duller, 2008) on stainless steel discs using silicone oil.




Fig. 1: (a) Location of the study area in South Africa. (b) Aerial photo map of the study area. (c-e) Photographs of the sampled sediment sections: Jojosi-Triple-Junction (c), Jojosi 1 (d) and Jojosi 5 (e).



## 2.2 Luminescence instrumentation and measurement conditions

### *Multi-grain measurements*

Luminescence measurements were performed on various Risø TL/OSL DA20 readers (Bøtter-Jensen et al., 2010), each equipped with a $^{90}$Sr/$^{90}$Y beta source and IR LEDs operating at 90 % power (~145 mW

cm$^{-2}$ for classic head at 100 %, ~300 mW cm$^{-2}$ for DASH at 100 %) at the CLL and at Risø (Technical University of Denmark, DTU). The beta sources were calibrated using Risø calibration quartz. The CLL calculates the dose rate of the instruments by fitting a regression line through multiple calibrations, with the instruments being calibrated every 6 months. The dose rate of the instruments at Risø are estimated by averaging the dose rate over the past six calibrations, with a calibration performed each month.

For multi-grain measurements the feldspar luminescence signal was detected through a combination of a 2 mm thick Schott BG39 filter and a 3 mm thick Corning 7-59 filter or BG3 filter, depending on the reader, allowing the transmission of the blue emission (~410 nm, Huntley et al., 1991). The single aliquot regenerative dose (SAR) protocol (Murray and Wintle, 2000) was adapted for feldspars as post-IR IRSL protocol (Thomsen et al., 2008). To decide on an appropriate measurement protocol, a dose recovery

preheat plateau, a residual preheat plateau and a fading preheat plateau test were performed on samples JOJO-85U and JOJO-1-2 (Fig. 2). For these plateau tests six different preheat and post-IR IRSL stimulation temperature combinations (post-IR IRSL temperature 25°C–30°C < preheat temperature, see Table S2 for details) were tested, using three aliquots per combination. For the dose recovery test and the residual dose measurements, the multi-grain aliquots were bleached in a Hönle Sol2 solar simulator for

24 hours. The multi-grain aliquots used in the dose-recovery test were given a dose of 100 Gy. For dose-recovery ratio calculations, the average residual dose obtained for each temperature combination was subtracted from the dose measured in the dose-recovery test. The dose-recovery ratio was then calculated by dividing the measured dose (residual corrected) by the given dose. Based on the results, the measurement protocol described in Table 1 was selected and used for equivalent dose measurements. The

chosen protocol was validated for all samples using a dose-recovery test and all samples showed dose-recovery ratios within 10 % of unity. Fading was also measured for all samples using the protocol outlined





in Table 1 and following the procedure by Auclair et al. (2003), with pauses of 0 s, 1000 s, 10,000 s, and 100,000 s inserted between steps 2 and 3. The pauses of 0 s and 10,000 s were repeated at the end of the fading sequence, to check for changes in sensitivity. The measurements were reproducible to ± 10 %.

Obtained fading rates are displayed in Fig. S3 for both luminescence signals investigated. The IRSL$_{50}$ signal shows fading rates ($g_{2days}$) ranging from 0.3 ± 1.5 %/decade to 6.2 ± 1.1 %/decade with an average fading rate of 2.9 ± 0.3 %/decade (n = 27). The post-IR IRSL$_{225}$ signal exhibits overall low fading rates, ranging from -0.6 ± -1.9 %/decade to 3 ± 2 %/decade, with an average (± standard error) fading rate of 1.6 ± 0.2 %/decade (n = 27). Obtained post-IR IRSL$_{225}$ ages were not corrected for fading (cf. Roberts,

190 2012).



**Fig. 2. Results of preheat plateau tests performed on samples JOJO-1-2 and JOJO-85U using multi grain aliquots. a and d) Dose-recovery preheat plateau test (sample-specific average residuals subtracted), b and e) Residual preheat plateau test, and c and f) fading preheat plateau test. Five temperatures were tested, with the preheat temperature being 20 °C, 25 °C or 30 °C > than the post-IRS IRSL temperature (see details in Table S2), depending on the protocol tested. The selected temperature combination is highlighted in grey, and the protocol is given in Table 1.**





Multi-grain aliquot $D_e$ measurements were made for all samples, and at least 36 aliquots were measured for each sample. To obtain a $D_e$ from the luminescence signal, we integrated the initial 10 s of the signal and subtracted the last 20 s as a background. Dose response curves were fitted using a single saturating exponential function and single aliquot equivalent doses were accepted with recycling ratios within 20 % of unity, the relative test dose error smaller than 20 %, and with a $T_n$ signal three standard deviations above the background. For SGC determination, in addition of these acceptance criteria, only aliquots which exhibited dose response curves with a reduced chi square value of the fit below 5 and a figure of merit of the fit below 15 % were accepted (see Li et al., 2018).

Table 1. Post-IR IRSL$_{225}$ measurement protocol used for feldspar measurements. Stimulation times were adjusted for SG measurements (given in parentheses) due to the increased power density of the stimulation light at the sample position (see text for details). [a]For $D_e$ measurements a test dose of 40 Gy was used for all samples, except for the two oldest samples (JOJO-TRPL-1 and JOJO-TRPL-2) where a test dose of 75 Gy was used.

| Step | Treatment | Observed |
|------|-----------|----------|
| 1 | Beta dose | |
| 2 | Preheat 250 °C, 60 s | |
| 3 | IRSL 50 °C, 200 s (2 s for SG) | |
| 4 | post-IR IRSL at 225 °C, 300 s (3 s for SG) | $L_x$ |
| 5 | Test dose[a] | |
| 6 | Preheat 250 °C, 60 s | |
| 7 | IRSL 50 °C, 200 s (2 s for SG) | |
| 8 | post-IR IRSL at 225 °C, 300 s (3 s for SG) | $T_x$ |

*Single grain measurements*

All single grain measurements were performed on the same Risø luminescence reader at the CLL, with the $^{90}Sr/^{90}Y$ source delivering a dose rate of ~0.074 Gy s$^{-1}$. For single grain $D_e$ determination, 12 to 25 single grain discs (1200-2500 grains) were measured for each sample following the protocol outlined in Table 1 and by stimulating the individual grains using an IR (830 nm) 140 mW Transistor-Transistor Logic modulated laser (Bøtter-Jensen et al., 2003; Duller et al., 2003). The stimulation time was adjusted for the single grain measurements (cf. Table 1). The feldspar luminescence signal was detected through a combination of a 2 mm thick Schott BG39 filter and a 3 mm thick Corning 7-59 filter, transmitting





around ~410 nm (Huntley et al., 1991). For $D_e$ determination the initial 0.2 s were used as signal, and the last 0.4 s were subtracted as background.

A single saturating exponential function was used to fit $L_x/T_x$ values obtained for single grains. Single grains were accepted when the relative test dose error was smaller than 20 %, and their $T_n$ signal three standard deviations above background. No recycling ratio could be obtained due to the spatial dose non-uniformity of the beta source and the resulting uneven delivery of beta radiation to the single grain discs, with a coefficient of variation in dose across the disc of 11.5 %. The doses administered were corrected

for this spatial dose non-uniformity of the beta source following Lapp et al. (2012) by using correction factors based on GAFChronomic Dosimetry Medium measurements and the correction software (CorrSGbin) provided by Risø. For the single grain SGC, additional rejection criteria (reduced chi square below 5 and figure of merit below 15, Li et al., 2018) were used.

### 2.3 Dose rate determination

To estimate the external dose rate, uranium (U), thorium (Th) and potassium (K) contents were determined by high-resolution gamma spectrometry. Approximately 200 g of dried, homogenised sediment was stored in an airtight box for at least four weeks to compensate for radon loss induced by sample preparation, before measurement with an Ortec Profile MSeries GEM Coaxial P-type high-precision Germanium Gamma-Ray detector.

The internal K-concentration was determined using a Risø GM beta multicounter system (Bøtter-Jensen and Mejdahl, 1985). The counting rates (48 h) of two subsamples of the K-feldspar separates of all nine samples were compared to the counting rates obtained for a K-feldspar standard (FK-N, Govindaraju, 1995) and based on these data the K- concentration of the samples was determined. The average ± standard deviation of the two measurements per sample (cf. Table 2) was used for internal dose rate

calculations. Since the obtained values are surprisingly low (<2 % $K_2O$), we furthermore performed single grain $K_2O$-concentration measurements using a JEOL JXA-8900RL Electron Microprobe housed at the Institute of Geology and Mineralogy (University of Cologne). To be able to determine the $K_2O$-concentration of individual grains and link it to the luminescence emitted by these grains, the grains from



a single grain disc were embedded in epoxy and polished following the procedure outlined by Maßon et al. (2024). A total of 46 grains were analysed for their mineral chemistry, the results of which are provided in Supplementary Tables 1a and 1b. The internal $K_2O$- concentration of luminescent grains (divided in grains emitting $IRSL_{50}$ only, post-IR $IRSL_{225}$ only, and those emitting both signals) can be found in Fig. S1 in the supplementary material. For alkali feldspars one would expect to see $K_2O$ contents ranging from 0 wt% to 16.9 wt%. Measurements of the $K_2O$-concentration of feldspar single grains in this work revealed a low average (± standard error, $n_{datapoints}$ = 135, $n_{grains}$ = 45) of 2.1 ± 0.4 wt%, despite this entire range being present in the data (min = 0, max = 16.3 wt%), 2.1 ± 0.4 wt%. Figure S1a also shows that $IRSL_{50}$-, post-IR $IRSL_{225}$- and non-emitting grains exhibit this range. From the 45 grains measured, 35 grains did not have a single measurement point with $K_2O$- concentration >3 wt%. Thus, despite the wide range of $K_2O$-concentrations found, the microprobe results support the low bulk K-contents determined using a beta counter, thus the results obtained through beta counting are used for dose rate calculations.

Dose rate and age calculations were performed using DRAC (Durcan et al., 2015). For samples JOJO-TRPL-2 and JOJO-TRPL-3 a user defined gamma dose rate was calculated using the *scale_GammaDose* function (Riedesel et al., 2023) available in the *R Luminescence* package to account for variations in gamma dose rate between layers influencing these two luminescence samples. To convert U, Th and K contents into dose rates, dose rate conversion factors of Guérin et al. (2011) were used. Alpha and beta grain size attenuation factors following Bell (1980) and Guérin et al. (2012) were applied, respectively. For coarse-grained feldspars, an alpha efficiency of 0.11 ± 0.03 (Balescu and Lamothe, 1993) was assumed. The cosmic dose rate was calculated according to Prescott and Hutton (1994). For all luminescence and dosimetry samples, the water content was determined by weighing the freshly collected wet sample, and comparing it to its weight after drying the sediment for two days at 45 °C. The average water content ± standard error of all measurements is 15 ± 2 %, however, to account for unknown but realistic fluctuations in water content in the past, a water content of 15 ± 5 % was used for all samples. Information on U, Th and K contents and total dose rates are given in Table 2.




**Table 2: Details regarding the samples, the measured radionuclide concentrations determined using gamma spectrometry, the internal K content measured using beta counting (see Bøtter-Jensen and Mejdahl, 1985) and the calculated environmental dose rates. The environmental dose rates were calculated using DRAC (Durcan et al., 2015). Depths are given as vertical depth from the surface of the gully walls, measured with a laser distometer. ªThe samples JOJO-TRPL-2AB, -2BE, and -3AB were only used to**
**calculate the gamma dose rate delivered to luminescence samples JOJO-TRPL-2 and JOJO-TRPL-3, thus no environmental dose rate was calculated for these samples.**

| Sample ID | Profile | Depth [m] | U [ppm] | Th [ppm] | K [%] | Internal K [%] | Environmental dose rate [Gy ka⁻¹] |
|---|---|---|---|---|---|---|---|
| *Profile Jojosi 1* | | | | | | | |
| JOJO-1-1 | Jojosi-1 | 3.65 | 0.48 ± 0.04 | 2.66 ± 0.19 | 0.298 ± 0.009 | 0.74 ± 0.06 | 0.74 ± 0.03 |
| JOJO-1-2 | Jojosi-1 | 2.00 | 0.42 ± 0.03 | 2.48 ± 0.18 | 0.288 ± 0.009 | 0.76 ± 0.13 | 0.74 ± 0.03 |
| JOJO-1-3 | Jojosi-1 | 0.65 | 0.46 ± 0.03 | 2.65 ± 0.19 | 0.280 ± 0.008 | 0.90 ± 0.02 | 0.81 ± 0.03 |
| *Profile Jojosi 5* | | | | | | | |
| JOJO-85U | Jojosi-5 | 2.00 | 0.39 ± 0.03 | 1.97 ± 0.14 | 0.283 ± 0.008 | 2.11 ± 0.09 | 0.80 ± 0.03 |
| JOJO-5-4 | Jojosi-5 | 0.60 | 0.39 ± 0.03 | 2.29 ± 0.17 | 0.290 ± 0.009 | 0.87 ± 0.03 | 0.78 ± 0.03 |
| JOJO-5-5 | Jojosi-5 | 0.35 | 0.41 ± 0.03 | 2.34 ± 0.17 | 0.282 ± 0.009 | 0.96 ± 0.07 | 0.81 ± 0.04 |
| *Profile Jojosi Triple-Junction* | | | | | | | |
| JOJO-TRPL-1 | Triple Junction | 1.20 | 0.41 ± 0.03 | 2.85 ± 0.20 | 0.445 ± 0.012 | 0.72 ± 0.02 | 0.92 ± 0.04 |
| JOJO-TRPL-2ABª | Triple Junction | Layer above JOJO-TRPL-2 | 0.33 ± 0.03 | 2.26 ± 0.16 | 0.242 ± 0.007 | NA | NA |
| JOJO-TRPL-2 | Triple Junction | 3.50, distance to top boundary: 7 cm, distance to lower boundary: 15 cm | 0.31 ± 0.02 | 1.49 ± 0.11 | 0.24 ± 0.01 | 1.15 ± 0.10 | 0.60 ± 0.03 |
| JOJO-TRPL-2BEª | Triple Junction | Layer below JOJO-TRPL-2 | 0.40 ± 0.03 | 2.77 ± 0.20 | 0.340 ± 0.010 | NA | NA |
| JOJO-TRPL-3ABª | Triple Junction | layer above JOJO-TRPL-3 | 0.48 ± 0.04 | 3.45 ± 0.25 | 0.365 ± 0.011 | NA | NA |
| JOJO-TRPL-3 | Triple Junction | 4.37, distance to layer boundary: 5 cm | 0.43 ± 0.03 | 2.68 ± 0.19 | 0.361 ± 0.011 | 1.71 ± 0.01 | 0.82 ± 0.04 |





## 3 Dose distributions and dose models in luminescence dating

Over the past three decades, different statistical models have been developed or were adapted for application to $D_e$ distributions (e.g. Galbraith et al., 1999; Bailey and Arnold, 2006; Arnold et al., 2009;
Guérin et al., 2017). The $D_e$ distribution measured for a sample is affected by a multicity of factors, for example, remnant doses in the grains, remaining due to incomplete resetting during previous transport, or post-depositional factors, such as beta dose heterogeneity. Depending on the shape of the obtained equivalent dose distribution, different dose models can and should be selected for dose calculations. Various attempts have been made to assist in deciding on the appropriate dose model for the obtained $D_e$
distribution (e.g. Bailey and Arnold, 2006; Galbraith and Roberts, 2012; Thomsen et al., 2016). However, the choice of an appropriate dose model remains challenging and complex, further complicated by the development of new statistical approaches for evaluating luminescence dose information (e.g. Philippe et al., 2019; Li et al., 2024).

Different dose models exist, which can be grouped in: (i) dose models which do not account for
uncertainties on individual equivalent doses when calculating a palaeodose (e.g. Clarke, 1996; Fuchs and Lang, 2001), (ii) dose models which incorporate the uncertainties (e.g. Galbraith et al., 1999; Guérin et al., 2017), and (iii) Bayesian models, which may include information beyond individual dose values and their uncertainties (e.g. Philippe et al., 2019; Li et al., 2024).

In the following, we will focus on the description and application of selected models from groups (ii) and
(iii); we will refer to the models selected from group (ii) as *frequentists approaches*, and to those from group (iii) as *Bayesian hierarchical approaches* (following the terminology introduced by Philippe et al., 2019).

There are many differences between these two approaches, however, for the focus of this work, the crucial difference is that *frequentist approaches* require the distribution to contain parameterised $D_e$ values in the
form x ± y, where both x and y are finite values. Conversely, the *Bayesian hierarchical approaches* mentioned above allow for the presence of non-parameterised $D_e$ values within the measured populations.



In this work, we will compare the results obtained by applying the CAM (Galbraith et al., 1999) and the ADM (Guérin et al., 2017), both *frequentist approaches*, to our data sets. Guérin et al. (2017) detail the differences between the CAM and the ADM, here we would just like to highlight some main differences

between the two models: (i) the CAM calculates the most representative dose for a given distribution by taking the median of a lognormal distribution (which is calculated as the weighted geometric mean), whereas the ADM calculates the weighted arithmetic mean of the same lognormal distribution; (ii) the CAM treats the overdispersion of a sample as measurement uncertainty (so that it is included in the weighting of each estimate), in contrast the ADM only treats the intrinsic overdispersion (i.e. the

dispersion arising in a dose-recovery test) as measurement uncertainty (so that extrinsic overdispersion is not included in the weight of $D_e$ estimates). The rationale behind the ADM is that extrinsic overdispersion is modelled as arising from dose rate variability, rather than an experimental factor. To be able to use *frequentists approaches*, single grains or multi-grain aliquots with normalised luminescence signals ($L_n/T_n$ values) and/or their uncertainties not intercepting the dose response curve need to be excluded,

because no finite dose and/or uncertainty values can be provided.

There exists the possibility of using *frequentists approaches*, such as the CAM, despite having $L_n/T_n$ values and/or their uncertainties above the maximum asymptote of the dose response curve ($I_{max}$): a standardised growth curve (SGC, Roberts and Duller, 2004; Li et al., 2015a,b), combined with a *frequentist approach* applied to a distribution of $L_n/T_n$ values (Li et al., 2017, 2020) instead of a

distribution of $D_e$ values might be a possibility. The SGC $L_n/T_n$ approach is based on the establishment of an SGC. A CAM is then applied to all $L_n/T_n$ values (renormalized according to the SGC) of grains/multi-grain aliquots that passed the acceptance criteria, and the obtained central $L_n/T_n$ (with uncertainties) is interpolated onto the SGC. Extrapolation from the SGC onto the dose axis informs on the dose of the sample. This way it is possible to include all $L_n/T_n$ values in the calculation, even those of

saturated grains. However, this can only be used, if a sufficient part of the $L_n/T_n$ distribution is below $I_{max}$. Otherwise, the resulting dose is either infinite or has an infinite upper uncertainty.

*Bayesian hierarchical approaches*, such as BayLum, have been developed to enable a more holistic view on luminescence dose and age calculations, and they furthermore enable the inclusion of grains/multi-





grain aliquots for dose calculations, which have $L_n/T_n$ values above $I_{max}$. Such aliquots are poorly

informative, yet their inclusion in the population of interest has been shown to greatly extend the range

of measurable doses (e.g., Heydari and Guérin, 2018; Arce-Chamorro et al., 2024). Indeed, in BayLum

all aliquots from one sample belong to one population with a common central dose; so even poorly

informative aliquots bear information.

For this study we decided to evaluate and compare the use of two *frequentists approaches* (CAM and

ADM), the SGC $L_n/T_n$ method combined with the CAM, and one Bayesian hierarchical model (BayLum).

## 4 Results

### 4.1 Single grain dose distributions

The geomorphological environment at Jojosi, was expected to result in partial bleaching of the

luminescence signals, likely due to short transport distances between source and sink, and due to rapid

erosion and transport. We thus considered single grain measurements as the most appropriate means of

measurement for the luminescence samples presented in this study. Contrary to quartz, a large portion

(~30-60 %) of individual grains of K-rich feldspar usually give detectable luminescence signals (e.g. Li

et al., 2011, Reimann et al., 2012), but unfortunately at Jojosi only $6.0 \pm 1.2$ % ($n_{measured} = 16,700$) of the

grains measured gave detectable luminescence signals in case of the post-IR IRSL$_{225}$ signal, and $20.4 \pm$

3.4 % in case of the IRSL$_{50}$ signal (measured as part of the post-IR IRSL$_{225}$ protocol (cf. Table 1). Table

3 shows the total number of grains measured, the number of accepted grains and the number of grains

where the natural signal appeared to be in saturation for the IRSL$_{50}$ and post-IR IRSL$_{225}$ signals,

respectively.

We determined single grain equivalent doses for all nine samples. To assess the appropriateness of our

measurement protocol, we also performed two single-grain dose recovery tests (samples JOJO-1-3 and

JOJO-85U) with a given dose of 100 Gy (see Fig. S6). The single-grain dose recovery distributions exhibit

overdispersion values of $20 \pm 2$ % ($n_{accepted} = 221$) and $12 \pm 4$ % ($n_{accepted} = 350$) for the IRSL$_{50}$ signal and

of $21 \pm 4$ % ($n_{accepted} = 58$) and $11 \pm 3$ % ($n_{accepted} = 74$) for the post-IR IRSL$_{225}$ signal, for JOJO-1-3 and



JOJO-85U, respectively. The relative overdispersion determined in these single grain dose-recovery tests

are lower than those obtained for the $D_e$ distributions, where the overdispersion ranges from 33 % to 56 % for the IRSL$_{50}$, and from 34 % to 75 % in case of the post-IR IRSL$_{225}$ signal (see Table 4). However, the natural dose distributions all appear log-normally distributed and do not show a prominent leading edge, as is expected for partially bleached samples (e.g. Reimann et al., 2012). Examples of the dose distributions obtained for one of the younger samples (JOJO-1-1) and the oldest sample (JOJO-TRPL-1)

are shown in Figure 3.

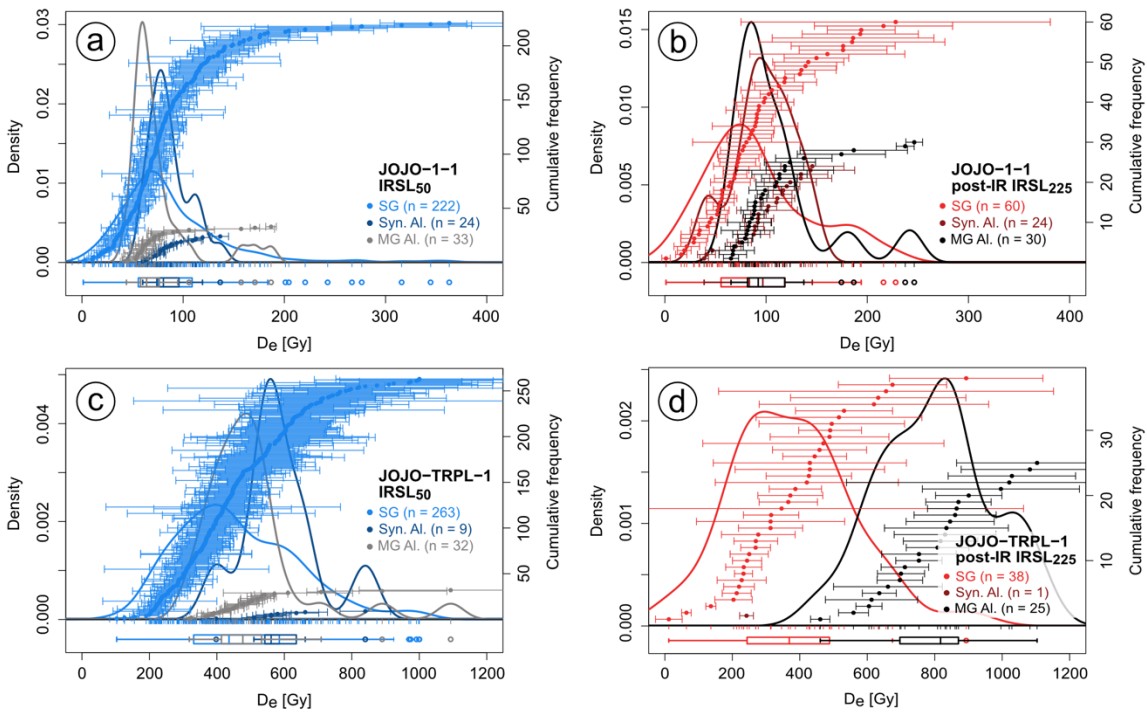

**Fig. 3. Kernel density plots (KDE) plots of equivalent dose distributions for the IRSL$_{50}$ (a, c) and post-IR IRSL$_{225}$ (b, d) signal for the youngest sample JOJO-1-1 (a, b) and the oldest sample JOJO-TRPL-1 (c, d). Only the number of accepted grains/aliquots is displayed in the figure and legend. Dose distributions for single grains (*SG*), synthetic aliquots (*Syn. Al.*) and multi-grain aliquots (*MG Al.*) are shown.**





**Table 3. Overview of single grains measured (*N*), accepted for equivalent dose calculations (*n*), and number of saturated grains (*n_sat*). Grains were accepted when the uncertainty on the natural test dose response ($T_n$) was less than 20 % and the $T_n$ signal three standard deviation above background. The relative number of accepted grains is calculated in relation to the total number of grains measured. Saturated grains are all included in the accepted grains and their percentage is calculated from the total number of accepted grains.**


| Sample ID | Luminescence signal | N | n (%) | $n_{sat}$ (%) |
|---|---|---|---|---|
| *Profile Jojosi 1* | | | | |
| **JOJO-1-1** | $IRSL_{50}$ | 2400 | 225 (9.4 %) | 3 (1.3 %) |
| | post-IR $IRSL_{225}$ | 2400 | 64 (2.7 %) | 4 (6.3 %) |
| **JOJO-1-2** | $IRSL_{50}$ | 2800 | 320 (11.4 %) | 4 (1.3 %) |
| | post-IR $IRSL_{225}$ | 2800 | 75 (2.7 %) | 6 (8.0 %) |
| **JOJO-1-3** | $IRSL_{50}$ | 2100 | 293 (14 %) | 2 (0.7 %) |
| | post-IR $IRSL_{225}$ | 2100 | 72 (3.4 %) | 6 (8.3 %) |
| *Profile Jojosi 5* | | | | |
| **JOJO-85U** | $IRSL_{50}$ | 1300 | 514 (41.6 %) | 4 (0.7 %) |
| | post-IR $IRSL_{225}$ | 1300 | 147 (11.3 %) | 24 (16.3 %) |
| **JOJO-5-4** | $IRSL_{50}$ | 1800 | 377 (20.9 %) | 28 (7.4 %) |
| | post-IR $IRSL_{225}$ | 1800 | 83 (4.6 %) | 7 (8.4 %) |
| **JOJO-5-5** | $IRSL_{50}$ | 1700 | 222 (13.1 %) | 2 (0.9 %) |
| | post-IR $IRSL_{225}$ | 1700 | 59 (3.5 %) | 5 (8.5 %) |
| *Profile Jojosi Triple-Junction* | | | | |
| **JOJO-TRPL-1** | $IRSL_{50}$ | 1800 | 424 (23.6%) | 153 (36.1 %) |
| | post-IR $IRSL_{225}$ | 1800 | 113 (6.3 %) | 76 (66.7 %) |
| **JOJO-TRPL-2** | $IRSL_{50}$ | 1600 | 321 (20.1 %) | 32 (10 %) |
| | post-IR $IRSL_{225}$ | 1600 | 65 (4.1 %) | 31 (47.7 %) |
| **JOJO-TRPL-3** | $IRSL_{50}$ | 1200 | 349 (29.1 %) | 3 (0.9 %) |
| | post-IR $IRSL_{225}$ | 1200 | 69 (5.8 %) | 9 (13.0 %) |

The single grain measurements of all samples show differences in the relative number of saturated grains. Here, we regard grains as *saturated* when the $L_n/T_n$ ratio and/or the sum of this ratio plus its uncertainty





does not intercept the dose response curve, thus lies above $I_{max}$ (e.g., Heydari and Guérin, 2018; Chapot
et al., 2022). All grains which yield a finite dose ± finite uncertainties are regarded as *not saturated* in
this study. In Table 3 the relative proportion of *saturated grains* is given as percentage of the total number
of accepted grains ranges ($n_{sat}$). It ranges from 0.9 % to 36 % for the $IRSL_{50}$ signal, and from 6.3 % to 67
% for the post-IR $IRSL_{225}$ signal (cf. Table 3). The relative number of saturated grains systematically
increases with single grain CAM dose, except for sample JOJO-TRPL-3, where the number of saturated
grains is too low to fit the trend of the other samples. Neither the geomorphological setting of the sample
(cf. Fig. 1c), nor the luminescence characteristics, such as the fading rate (Fig. S3b) or the curvature
(expressed as $D_0$, derived from the single saturation exponential fit of the dose response curve using: $y =$
$a\left(1 - e^{-\frac{x+c}{D_0}}\right)$) of the dose-response curve (Fig. S7a) indicate any differences to the other samples.

Buylaert et al. (2013) have shown that comparing doses obtained for two luminescence signals, which
reset at different rates during sunlight exposure, can inform on how well the samples had been bleached
prior to burial. We make a similar comparison of single grain equivalent doses obtained for the $IRSL_{50}$
and post-IR $IRSL_{225}$ signals (Fig. 4). Figure 4a shows scattering of the doses around the 1:1 line, with an
average $IRSL_{50}$/post-IR $IRSL_{225}$ $D_e$ ratio of 0.89 ± 0.02. The ratio of $IRSL_{50}$ $D_e$/post-IR $IRSL_{225}$ $D_e$ (Fig.
4c) decreases with increasing dose. However, due to only few data points available for high post-IR
$IRSL_{225}$ doses, this trend is not significant, and we thus do not display the fit in Fig. 4c. However, if
uncertainties on the ratios are considered than 25 % of the ratios <1, agree with 1 within 1σ. 15 % of the
ratios >1 agree with 1 within 1σ. From fading tests performed on all samples (cf. Fig. S3b) we know that
the $IRSL_{50}$ fades more compared to the post-IR $IRSL_{225}$ signal. We thus interpret the underestimation of
some of the $IRSL_{50}$ doses as indicator for increasing fading, rather than incomplete bleaching of the post-
IR $IRSL_{225}$ signal.




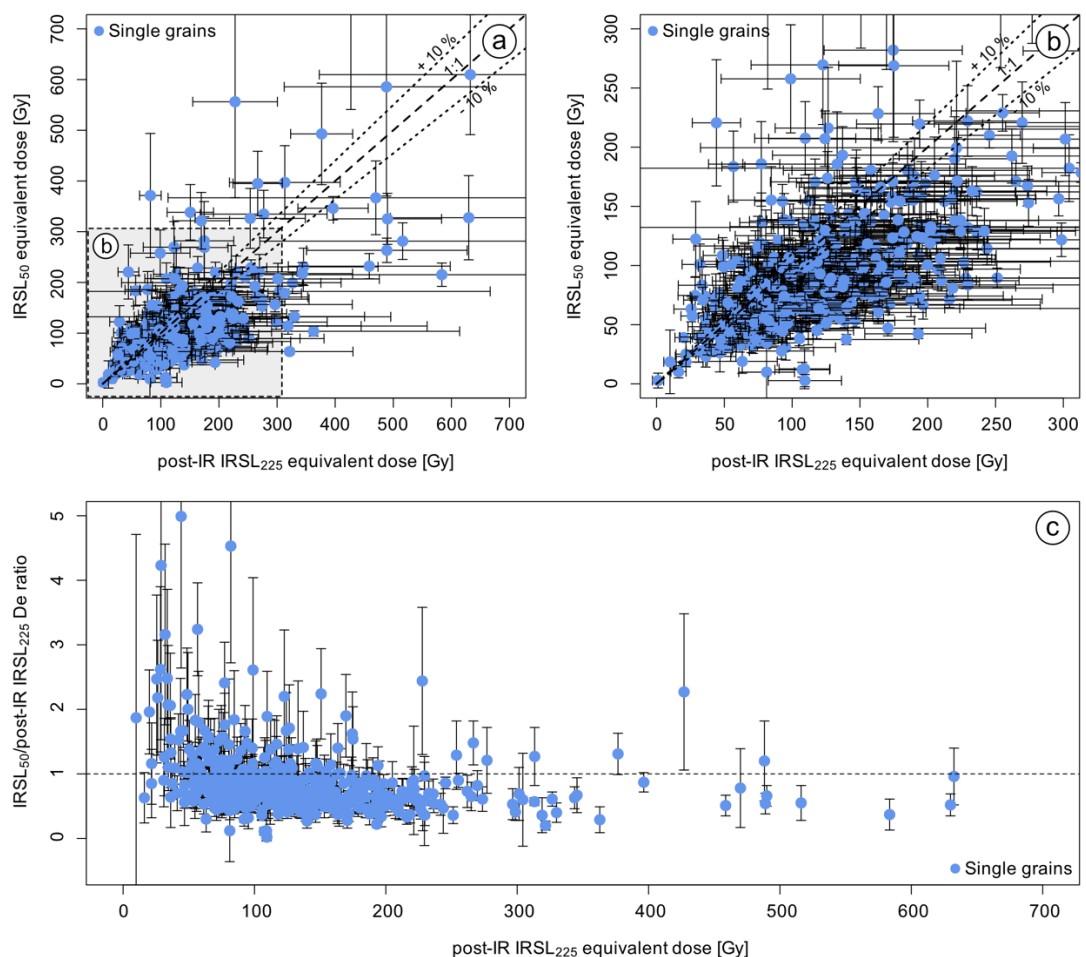

**Fig. 4. (a) Scatter plot of post-IR IRSL$_{225}$ doses compared to IRSL$_{50}$ doses obtained for single grains. Here the results of single grains of all samples are shown, which passed the acceptance criteria in case of both signals, and which yielded finite equivalent doses. The shaded rectangle shows the region of the plot highlighted in (b). (c) Ratios of IRSL$_{50}$ doses divided by post-IR IRSL$_{225}$ doses dependent on the size of the post-IRIRSL$_{225}$ dose calculated for all data points displayed in (a).**

## 4.2 Multi-grain aliquot dose distributions

The low yield of luminescent single grains (Table 3) makes single grain measurements very time- and labour intensive. Since no incomplete resetting could be detected on the single grain level, we tested the applicability of measuring multi-grain aliquots of all nine samples. Measuring multi-grain aliquots should decrease the time needed to acquire a sufficiently large data set. Multi-grain D$_e$ distributions are shown





alongside single grain $D_e$ distributions in Fig. 3, for samples JOJO-1-1 and JOJO-TRPL-1, for the $IRSL_{50}$ and post-IR $IRSL_{225}$ signal, respectively.

Similarly to the distributions obtained for single grains, the multi-grain distributions appear to be log-normal distributed, with a few outliers, and no prominent leading edge. The relative overdispersion ranges from $12 \pm 2$ % (JOJO-1-2) to $47 \pm 6$ % (JOJO-TRPL-2) for the $IRSL_{50}$ signal of multi-grain aliquots, from $14 \pm 5$ % (JOJO-85U) to $40 \pm 8$ % (JOJO-5-5) for the post-IR $IRSL_{225}$ signal (cf. Table 4). As expected, the multi-grain data show less scatter compared to the single grain data sets, indicating the effect of averaging of the luminescence signals. Interestingly, only four of the nine samples measured exhibited *saturated* multi-grain aliquots for the post-IR $IRSL_{225}$ signal (JOJO-85U, JOJO-TRPL-1, -2, -3), whilst the single grain data sets of all samples showed *saturated grains*.

### 4.3 Synthetic aliquot dose distributions

To further investigate the differences in relative saturation of the single grain and multi-grain aliquot data, we used the possibility of summing the luminescence of all single grains on a single grain disc to create synthetic aliquots. This was done by using the "*sum all grains*" function in Analyst (Duller, 2015). Thus, we accumulated synthetic aliquot data for 12 to 28 discs, with the exact number of measured discs per sample given in Table 3 and results given in Table 4. It should be noted that the stimulation time and intensity differ for these measurements, compared to multi-grain aliquot measurements. Furthermore, we here summed the luminescence signal of potentially 100 grains, whereas for small multi-grain aliquots we summed the signal of approximately 30 grains (Duller, 2008). The results are displayed for two samples in Fig. 3. Relative overdispersion values range from $13 \pm 3$ % (JOJO-1-3) to $24 \pm 5$ % (JOJO-85U) for the $IRSL_{50}$ signal, and from $14 \pm 3$ % (JOJO-1-3) to $47 \pm 6$ % (JOJO-TRPL-2) for the post-IR $IRSL_{225}$ signal. Note: no synthetic aliquot data is available for the post-IR $IRSL_{225}$ signal of sample JOJO-TRPL-1, because only a single synthetic aliquot passed the acceptance criteria while also giving a finite $L_n/T_n$ value and dose.



Due to the difference in number of samples exhibiting saturation on the single grain level compared to the multi-grain level, we also checked the synthetic aliquot data sets for saturation. Here five samples contained saturated aliquots. This includes sample JOJO-5-4 and all four samples which also showed saturated multi-grain aliquots. As with single grains, JOJO-TRPL-1 contains the most saturated multi-grain and synthetic aliquots. For four samples (JOJO-1-1, -2, -3, and JOJO-5-5), it is interesting to note
that despite containing saturated grains, all synthetic aliquots gave finite equivalent doses.

## 4.4 Palaeodose calculations

Due to the large range of equivalent doses and various amounts of *saturated grains*/multi-grain *aliquots* obtained for the different samples, we tested up to four different dose models on the different data sets (single grains, synthetic aliquots and multi-grain aliquots). We will focus on the post-IR IRSL$_{225}$ signal
only. The different dose models will not be evaluated for the IRSL$_{50}$ signal: primarily because the IRSL$_{50}$ signal exhibits a larger signal loss due to fading and we thus considered the dose model evaluation to be more robust for a non-fading luminescence signal. Furthermore, our comparison of IRSL$_{50}$ and post-IR IRSL$_{225}$ doses revealed that the post-IR IRSL$_{225}$ signal had been reset sufficiently prior to burial. To further support this, IRSL$_{50}$ ages were calculated using BayLum and are presented in the supplementary
material to guide in the evaluation of resetting of the luminescence by sunlight exposure at the time of sediment deposition (cf. Fig. S10).

Details regarding the use of the four selected dose models can be found in the supplementary material (i.e., the size of the intrinsic over-dispersion σ$_m$ used in ADM calculations, convergence criteria for BayLum and SGC validation). For the synthetic aliquots, only the *frequentist approaches* (CAM and
ADM) were tested. The results of the different dose model calculations for the three data sets investigated are given in Table 4. Besides information on the calculated dose (± a confidence interval of 68 % or a credible interval at 68 %), the table also lists relative overdispersion values as well as the number of grains and multi-grain, as well as synthetic aliquots included in the calculation (*n*). Figure 5 visualises the comparison of the different dose models applied to the three datasets.



Comparing CAM and ADM for multi-grain and single grain data sets, as well as for synthetic aliquots, reveals good agreement between these two frequentist approaches for the entire dose range from ~80 to ~800 Gy (Fig. 5a, b). For the multi-grain data set the ratio of CAM/ADM is $0.96 \pm 0.02$ (n = 9), for synthetic aliquots $0.97 \pm 0.01$ (n = 8), and for single grains $0.92 \pm 0.02$ (n = 9), with JOJO-5-5 ($0.77 \pm 0.11$) causing the underestimation of this ratio due to a few grains with very high doses. These ratios

indicate CAM doses smaller than ADM doses, with the ratio being smallest for the single grain dataset. This systematic difference is expected because the average of a lognormal distribution is always greater than (or equal to) its median; and the difference between these values increases when the dispersion increases.

If we compare the single grain ADM with multi-grain ADM, we find that the single grain doses

underestimate the multi-grain doses by about 15 % (ratio of $0.83 \pm 0.07$, n = 9). Similar underestimations have been reported for quartz measurements (e.g., Guérin et al., 2015; Thomsen et al., 2016; Singh et al., 2017). For quartz it has been shown that this underestimation can at least partly be caused by saturation effects and can be significantly reduced by excluding grains not able to measure the absorbed dose accurately (i.e., grains with $D_0$ values less that the absorbed dose, e.g., Thomsen et al., 2016). We also

tested if excluding grains of certain $D_0$ thresholds has an effect on the CAM and ADM doses for the single grain data set of all samples, but we could not find any effect on the doses calculated, despite $D_0$ threshold filtering resulting in the exclusion of some of the saturated grains (data not shown). Since the $D_0$ filter had no effect on the CAM and ADM doses, we did not apply this additional rejection criterion. The ratio between the synthetic aliquot ADM doses and the multi-grain ADM doses is $0.95 \pm 0.05$ (n = 8, no

synthetic aliquot data could be generated for sample JOJO-TRPL-1, Table 4)

Comparing ADM doses to BayLum doses shows that when using multi-grain aliquots, BayLum and ADM results agree on average (BayLum/ADM ratio = $0.99 \pm 0.03$, n = 9), despite a large deviation in BayLum/ADM ratio in case of JOJO-TRPL-1 ($0.75 \pm 0.04$). For single grains, ADM doses systematically underestimate BayLum doses ($1.23 \pm 0.10$, n = 9), with ADM and BayLum doses being consistent until

the proportion of saturated grains exceeds ~9 %. If we use the SGC $L_n T_n$ approach on the multi-grain data




set and compare it to the ADM results, the ratio to ADM doses is $0.89 \pm 0.03$ (n = 9). For single grains, the SGC $L_nT_n$ to ADM ratio is $1.15 \pm 0.13$ (n = 9). Especially large deviations are visible for samples JOJO-5-4, JOJO-TRPL-1 and JOJO-TRPL-2.

BayLum and the SGC $L_nT_n$ approach both allow for the inclusion of saturated grains. Comparing these

two models (Fig. 5g, h) reveals SGC $L_nT_n$/BayLum ratios of $1.11 \pm 0.06$ (n = 9) and $1.01 \pm 0.05$ (n = 9) for single grains and multi-grain aliquots, respectively. The deviation is larger for single grains, largely because of sample JOJO-TRPL-1. For this sample, no uncertainties could be calculated for the SGC $L_nT_n$ approach due to the CAM $L_nT_n$ uncertainties not intercepting the SGC. For multi-grain aliquots these two models yield relatively consistent results, with JOJO-TRPl-1 showing the largest deviation with a ratio of

$0.72 \pm 0.07$.

On average, for multi-grain aliquots, all dose models result in burial doses in agreement with each other, indicating the suitability of all dose models tested. For single grains, however, the relative number of saturated grains influences the decision for the appropriate dose model. Larger deviations between the models are observed for the higher dose samples, with JOJO-TRPL (containing 67 % saturated grains),

showing the highest deviation in most cases.







**Fig. 5. Comparison of the different dose models and their results for single grains, multi-grain aliquots, and synthetic aliquots. For each dose model comparison the doses resulting from the calculations are visualised as scatter plot of the doses and as ratios dependent on the ADM dose (c) or the percentage of grains in saturation (d,g,h).**




**Table 4. Results of post-IR IRSL$_{225}$ D$_e$ determination and dose modelling for single grains (SG), synthetic aliquots (SynAl) and multi-grain aliquots (MG). OD = overdispersion. $^a$No reliable overdispersion could be calculated using the calc_CentralDose() function in the RLuminescence package due to the low number of accepted finite synthetic aliquots.**

| Sample ID | Aliquot size | n for CAM, ADM/BayLum/ SGC L$_n$T$_n$ | CAM [Gy] | Rel. OD [%] | ADM [Gy] | BayLum D$_e$ [Gy] | BayLum 1σ range | SGC L$_n$T$_n$ (CAM) [Gy] |
|---|---|---|---|---|---|---|---|---|
| *Profile Jojosi 1* | | | | | | | | |
| JOJO-1-1 | SG | 60/64/37 | 81 ± 6 | 45 ± 6 | 88 ± 6 | 96 | 88 - 101 | 94 ± 10 |
| | SynAl | 24/24/- | 91 ± 6 | 31 ± 6 | 95 ± 5 | NA | NA | NA |
| | MG | 30/30/34 | 102 ± 7 | 33 ± 5 | 107 ± 9 | 112 | 101 - 119 | 96 ± 6 |
| JOJO-1-2 | SG | 69/75/33 | 89 ± 5 | 41 ± 5 | 95 ± 5 | 104 | 97.3 - 109 | 92 ± 10 |
| | SynAl | 25/28/- | 91 ± 6 | 28 ± 5 | 93 ± 6 | NA | NA | NA |
| | MG | 35/35/18 | 93 ± 3 | 14 ± 2 | 93 ± 2 | 96.0 | 94 – 99 | 106 ± 6 |
| JOJO-1-3 | SG | 66/72/33 | 98 ± 6 | 37 ± 5 | 103 ± 5 | 109 | 102 - 114 | 93 ± 9 |
| | SynAl | 21/21/- | 93 ± 5 | 22 ± 5 | 94 ± 5 | NA | NA | NA |
| | MG | 28/28/34 | 92 ± 3 | 14 ± 3 | 92 ± 4 | 89.5 | 85.3 – 91.3 | 95 ± 6 |
| *Profile Jojosi 5* | | | | | | | | |
| JOJO-85U | SG | 122/147/65 | 115 ± 5 | 34 ± 4 | 120 ± 5 | 157 | 152 - 163 | 124 ± 10 |
| | SynAl | 10/13/- | 128 ± 7 | 14 ± 5 | 128 ± 7 | NA | NA | NA |
| | MG | 33/35/6 | 143 ± 15 | 59 ± 7 | 168 ± 15 | 180 | 163 - 197 | 152 ± 26 |
| JOJO-5-4 | SG | 76/83/39 | 104 ± 6 | 45 ± 5 | 114 ± 9 | 118 | 112 - 125 | 95 ± 12 |
| | SynAl | 16/18/- | 111 ± 9 | 29 ± 6 | 115 ± 8 | NA | NA | NA |
| | MG | 34/34/13 | 129 ± 6 | 25 ± 3 | 132 ± 8 | 135 | 125 - 137 | 121 ± 7 |
| JOJO-5-5 | SG | 54/59/43 | 88 ± 9 | 74 ± 8 | 115 ± 12 | 120 | 110 - 129 | 110 ± 12 |
| | SynAl | 17/17/- | 113 ± 12 | 40 ± 8 | 122 ± 13 | NA | NA | NA |
| | MG | 28/28/25 | 117 ± 5 | 23 ± 3 | 119 ± 6 | 119 | 114 - 125 | 131 ± 9 |
| *Profile Jojosi Triple-Junction* | | | | | | | | |
| JOJO-TRPL-1 | SG | 38/113/40 | 325 ± 29 | 45 ± 8 | 356 ± 31 | 501 | 470 - 523 | 746 ± NA |
| | SynAl | NA/18/- | NA | NA$^a$ | NA | NA | NA | NA |
| | MG | 25/39/36 | 752 ± 34 | 18 ± 4 | 756 ± 31 | 568 | 546 - 581 | 785 ± 76 |
| JOJO-TRPL-2 | SG | 31/65/25 | 182 ± 13 | 26 ± 7 | 186 ± 13 | 361 | 323 - 389 | 269 ± 56 |
| | SynAl | 6/16/- | 358 ± 37 | NA$^a$ | 358 ± 11 | NA | NA | NA |
| | MG | 34/35/18 | 262 ± 22 | 47 ± 6 | 291 ± 21 | 309 | 286 - 330 | 289 ± 25 |
| JOJO-TRPL-3 | SG | 59/69/32 | 178 ± 10 | 34 ± 5 | 186 ± 12 | 205 | 195 - 218 | 190 ± 18 |
| | SynAl | 10/12/- | 176 ± 6 | NA$^a$ | 178 ± 6 | NA | NA | NA |
| | MG | 31/34/42 | 217 ± 12 | 25 ± 4 | 222 ± 14 | 212 | 202 - 224 | 200 ± 16 |

## 5 Discussion

For the present study constraining the depositional ages of sediment accretion exposed in the Jojosi Donga, a robust chronology is of importance for an archaeological interpretation of the artefacts found at Jojosi, as well as to contextualise the geomorphological and palaeoclimatic conditions under which



successive phases of gully cut-and-fill occurred, leading to the sheetwash colluvium-palaeosol succession
exposed in today's dendritic donga landscape. To ensure the establishment of a robust luminescence-based chronology for Jojosi, we here evaluate the data sets generated for the post-IR IRSL$_{225}$ signal regarding their performance, including the luminescence signal saturation level, the dose model used for burial dose calculations (section 5.1), as well as the derived ages (section 5.2).

**5.1 Luminescence signal saturation and dose model evaluation**

When comparing the single grain-based data set to the multi-grain data set, two interesting observations can been made: (1) Whilst the post-IR IRSL$_{225}$ single grain data sets contains 6.3 % to 67 % saturated grains (cf. Table 3, section 4.1), the multi-grain aliquot data sets only show saturated aliquots for four (samples JOJO-85U, JOJO-TRPL-1,-2,-3) out of the nine samples. The number of saturated aliquots is greater in the case of synthetic aliquots than for multi-grain aliquots, with five out of nine samples
showing saturated synthetic aliquots. (2) Despite some (systematic) differences, on average, a comparison of burial doses calculated using the different models for single grains and multi-grain as well as synthetic aliquots revealed good agreement between the dose models tested, as well as between the single grain and multi-grain aliquot-based data.

Measuring single grains is more time- and labour consuming, especially for samples with a low yield of
luminescent grains, as is the case for Jojosi. In this case, measuring single grains instead of multi-grain aliquots would only be advantageous if it would increase the accuracy and precision of the luminescence ages calculated. The question whether it is truly necessary to measure single grains has been asked for quartz-based luminescence measurements, and discussions have sprouted from this (cf. Thomsen et al., 2016; Feathers, 2017; Thomsen et al., 2017). Different studies have evaluated the use of single grains
compared to aliquots for quartz (e.g. Thomsen et al., 2016; Colarossi et al., 2020) and feldspars (e.g. Sutikna et al., 2016; Guo et al., 2020), respectively. Nevertheless, this remains a challenging discussion, particularly complicated due to site-specific sample characteristics, such as incomplete bleaching or post-depositional mixing (e.g. Jacobs et al., 2008; van der Meij et al., under review). Furthermore, methodical





questions remain, for example regarding the higher scatter in single grain data sets (e.g. Thomsen et al.,

2005; Autzen et al., 2017; Hansen et al., 2018).

To investigate why the multi-grain data is less affected by saturation compared to the single grain (and the synthetic aliquot) data, we compared the dose response curves of all single grains, multi-grain and synthetic aliquots measured, including their $D_0$ values, which are a measure of the curvature of the dose response curve. Wintle and Murray (2006) proposed the use of $2*D_0$ as a maximum reliability threshold

for quartz OSL dating. Although we did not apply this criterion to our data set, comparing $D_0$ values for the different dose response curves of individual multi-grain aliquots and grains can give information on the curvature and thus saturation dose of the different curves. Fig. S7a shows the $D_0$ values of the post-IR IRSL$_{225}$ signal of the three data sets for all nine samples as boxplots, and Fig. S7b shows the fitted dose response curves. From Fig. S7a it becomes evident that despite large apparent scatter in $D_0$ values

for single grain dose response curves, $D_0$ tends to be lower for single grains and synthetic aliquots compared to the multi-grain aliquots, with the medians of 200 Gy for single grains (n = 742), 185 Gy for synthetic aliquots (n = 160), and 266 Gy for multi-grain aliquots (n = 263). Similarities in $D_0$ between single grains and synthetic aliquots could arise from the same signals being used for dose response curve construction. In case of synthetic aliquots, the luminescence emitted by single grains (placed in holes on

single grain discs) in response to IR laser stimulation is summed. Contrastingly, the multi-grain aliquot data was obtained by using infrared LEDs and the multi-grain aliquots were stimulated for 300 s in case of the post-IR IRSL$_{225}$ signal (compared to 3 s for single grains). Further explanations could be (i) differences in wavelength and power density of the excitation light sources (Classic head: $870 \pm 40$ nm, ~145 mW cm$^{-2}$; DASH: $850 \pm 33$ nm >300 mW cm$^{-2}$) and laser stimulation (830 nm, 140 mW), (ii) the

contribution of weakly luminescent grains to multi-grain aliquots, (iii) differences in signal integration and potential effects of this on growth curve shape.

Despite no explanation for the difference in $D_0$ between the different data sets can be given here, we identify the difference in $D_0$ for the different data sets as a plausible explanation for the difference in saturated grains and multi-grain aliquots between the different data sets. Higher $D_0$ values, in combination

with a greater number of finite $L_nT_n$ values results in the inclusion of these $L_nT_n$ values (and their finite





equivalent doses) in frequentist approaches such as the ADM and CAM, i.e. the average ratio for all samples of single grain ADM to multi-grain aliquot ADM is $0.83 \pm 0.07$ (n = 9), and it decreases with increasing dose (i.e., the ratio is $0.71 \pm 0.07$, n = 5, when only including samples with multi-grain aliquot doses greater than 120 Gy). BayLum and the SGC $L_nT_n$ method instead take the saturated grains and

multi-grain aliquots into account, thus causing BayLum and SGC $L_nT_n$ burial doses to be more consistent, i.e., the ratio of single grain BayLum to multi-grain ADM doses is $0.98 \pm 0.06$ (n = 9) and the single grain SGC $L_nT_n$ to multi-grain ADM doses is $0.89 \pm 0.03$ (n = 9). Note that in the case of BayLum, the saturated grains and multi-grain aliquots are not treated as finite values, but as an indication of the presence of a population with dose values greater than the maximum asymptote of the dose response curve (Heydari

and Guérin, 2018; Arce-Chamorro and Guérin, 2024, see also their supplementary material).

When comparing palaeodoses calculated using *frequentist approaches* (ADM and CAM) to doses obtained using BayLum (cf. Fig. 5b), a good agreement was found for most samples investigated in the case of single grains and multi-grain aliquots. However, for some samples in the single grain data sets, BayLum yields significantly larger doses compared to the ADM. Calculating a ratio of ADM dose and

BayLum dose and plotting it against the relative proportion of saturated grains (cf. Fig. 5d) reveals that the proportion of saturated grains seems to dictate the consistency between the two dose models for the single grain data set. If less than 15 % of the total grains measured are saturated, then ADM doses underestimate BayLum doses by a maximum of 10 %. However, if more than 15 % of the grains are saturated, then ADM doses underestimate BayLum doses for the respective samples by 25 % to 50 % (cf.

Fig. 5d). More than 15 % saturated grains can be found in samples with ADM palaeodoses >120 Gy and BayLum palaeodoses >160 Gy. Interestingly, whilst for single grains, BayLum doses are consistently higher than ADM doses for samples with more than 15 % saturated grains, for multi-grain aliquots the only sample showing a deviation between these two models is JOJO-TRPL-1. In the case of this sample, the ADM dose is larger than the BayLum dose. A cause of this underestimation of the BayLum dose

compared to the ADM dose could be that the model used in BayLum for dose and age calculation (function *Model_Age* in the R package BayLum, Philippe et al., 2019), is based on quartz data sets and uses a $D_0$ value of 50 Gy as starting point. As shown in Fig. 7a, multi-grain aliquots of feldspars from





Jojosi have a median $D_0$ of 266 Gy, thus much higher than $D_0$ values usually obtained for quartz. This indicates that BayLum might need to be adjusted for its use on high dose and near saturation feldspar

samples.

When comparing palaeodoses calculated using the ADM and the SGC $L_nT_n$ method, large deviations can be found for single grain results of the two samples with the highest doses and the largest relative number of saturated grains (JOJO-TRPL-1 and -2), indicating again an effect of the number of saturated grains on the calculated single grain burial doses. For the multi-grain data set, the SGC $L_nT_n$ /ADM ratios are

consistent. Larger $D_0$ values and finite multi-grain aliquots results, again, show that the choice of dose model and data set can influence the final burial dose, dependent on the relative number of saturated grains in the data set.

### 5.2 Age calculation and implications

Due to the large number of saturated grains present in some of the samples, we would like to include the

information they might be able to provide in our age calculations. Both BayLum and the SGC $L_nT_n$ approach allow us to do so, and we have shown that both methods yield consistent results. However, BayLum also, allows us to include stratigraphic information in the age calculations (Guérin et al., 2020, 2023; Heydari et al., 2020). Furthermore, the SGC $L_nT_n$ single grain results for JOJO-TRPL-1 did not yield finite uncertainties. Despite not considering the SGC $L_nT_n$ results further in this paper, the

comparison of the different methods has shown that the SGC $L_nT_n$ method performs well for feldspar single grain and aliquot data up to >700 Gy, supporting findings by Li et al. (2020).

Using BayLum with stratigraphic constraints, we were able to calculate the depositional ages of all nine sediment samples from Jojosi, using single grains and multi-grain aliquots. The results of the age calculations are given in Table 5 and are displayed as age-depth plots in Figure 6.

Single grain and multi-grain aliquots based post-IR IRSL$_{225}$ BayLum ages give consistent results within $1\sigma$, except for JOJO-1-3, where single grain and multi-grain aliquot results are consistent within $2\sigma$. From a practical point of view, multi-grain measurements are favoured for future work at Jojosi, due to greater





time-efficiency of the measurements, the higher intensity of the luminescence signals, and the larger saturation threshold.

The youngest age, ranging from 106 ka to 117 ka (1σ), could be obtained for the multi-grain aliquot data of JOJO-1-3. JOJO-TRPL-1 was identified as the oldest sample with the multi-grain aliquot age range spanning from 583 ka to 654 ka (1σ). With the calculated post-IR IRSL$_{225}$ ages, we show that erosional and depositional processes, leading to donga formation at Jojosi, took place at least from marine isotope stadium (MIS) 15 to MIS 5, thus spanning more than 500,000 years of Quaternary history in South Africa.

The post-IR IRSL$_{225}$ chronology of donga formation is the first of its kind for this area of South Africa, indicating donga formation already during the Middle Pleistocene. Clarke et al. (2003) combined IRSL$_{50}$ measurements with radiocarbon and found good agreement between these two chronometers, constraining colluvial sedimentation at Voordrag in KwaZulu-Natal over the past 100 ka. Clarke et al. (2003) interpreted the absence of older donga deposits with erosion during MIS 5e. Colarossi et al. (2020) used

paired quartz and feldspar single grain and multi-grain dating and revised the chronology by Clarke et al. (2003) and showed that the IRSL$_{50}$ by Clarke et al. (2003) suffered from underestimation due to fading. The chronology of the Voordrag site by Colarossi et al. (2020) shows single grain quartz and feldspar ages in agreement, up to ~40 ka. Above this threshold the quartz ages underestimate feldspar ages due to saturation of the quartz OSL signal, but the single grain post-IR IRSL$_{225}$ ages constrain donga deposition

up to ~110 ka. Similar time spans for colluvial deposition have been constrained by Wintle et al. (1993, 1995). These earlier studies used a combination of TL and IRSL measurements, both methods having disadvantages over modern post-IR IRSL measurements (i.e. hard to bleach signal and fading), but interestingly, these studies also constrained colluvial sedimentation over the late Pleistocene. The sediments at Jojosi, thus seem to be an exception for the preservation of colluvial sediments older than

100 ka, offering rare insights into Middle Pleistocene geomorphological processes.





**Table 5. Results of age calculations using BayLum for single grains and multi-grain aliquots.**

| Sample ID | Depth | SG/MG | BayLum Age [ka] | BayLum Age range 1σ [ka] | BayLum Age range 2σ [ka] |
|---|---|---|---|---|---|
| *Profile Jojosi 1* | | | | | |
| **JOJO-1-1** | 3.65 | SG | 145 | 137-152 | 130-161 |
| | | MG | 156 | 141-166 | 132-181 |
| **JOJO-1-2** | 2.00 | SG | 138 | 131-144 | 126-152 |
| | | MG | 132 | 121-139 | 115-150 |
| **JOJO-1-3** | 0.65 | SG | 130 | 123-137 | 117-143 |
| | | MG | 111 | 106-117 | 101-122 |
| *Profile Jojosi 5* | | | | | |
| **JOJO-85U** | 2.00 | SG | 205 | 191-217 | 180-231 |
| | | MG | 218 | 203-242 | 186-250 |
| **JOJO-5-4** | 0.60 | SG | 157 | 147-165 | 139-176 |
| | | MG | 175 | 160-187 | 149-202 |
| **JOJO-5-5** | 0.35 | SG | 143 | 132-153 | 123-162 |
| | | MG | 148 | 136-160 | 125-171 |
| *Profile Jojosi-Triple-Junction* | | | | | |
| **JOJO-TRPL-1** | 1.20 | SG | 582 | 541-615 | 511-653 |
| | | MG | 622 | 583-654 | 562-692 |
| **JOJO-TRPL-2** | 3.50 | SG | 545 | 505-583 | 467-622 |
| | | MG | 526 | 479-561 | 450-594 |
| **JOJO-TRPL-3** | 4.37 | SG | 251 | 231-269 | 215-289 |
| | | MG | 256 | 236-273 | 222-293 |

With the help of the luminescence ages presented here we can constrain the temporal context of the

archaeological sites interbedded within sections Jojosi 1 (excavated during the early 1990s, see Möller et al., in prep) and Jojosi 5 (excavated in 2023, see Will et al., 2024). Whilst it is straightforward to bracket the artefact lens in Jojosi 5, due to luminescence sampling taking place at the same time as the archaeological excavation, no exact artefact horizon could be determined and bracketed for Jojosi-1 due to the artefacts having been excavated in the early 1990s. For Jojosi 5 we can constrain modern human

activities between 136 ka to 187 ka (1σ, multi-grain aliquots), so at the end of the Middle Pleistocene and well within MIS 6. Although no absolute depth can be obtained for Jojosi 1, the artefact lens was likely once located between the samples JOJO-1-2 and JOJO-1-3 based on past photographic evidence and site reconnaissance in 2023 and thus to approximately 106 ka to 139 ka (Möller et al., in prep). Comparing the luminescence ages for the samples bracketing the artefact horizon in Jojosi 1 and Jojosi 5 reveals




diachronous human activities associated with discrete sedimentary and gully cut-and-fill context at these two closely spaced sites within the Jojosi Dongas. Further archaeological sites have been discovered and more research, including luminescence dating, will reveal the chronological extent of past human activities at Jojosi.

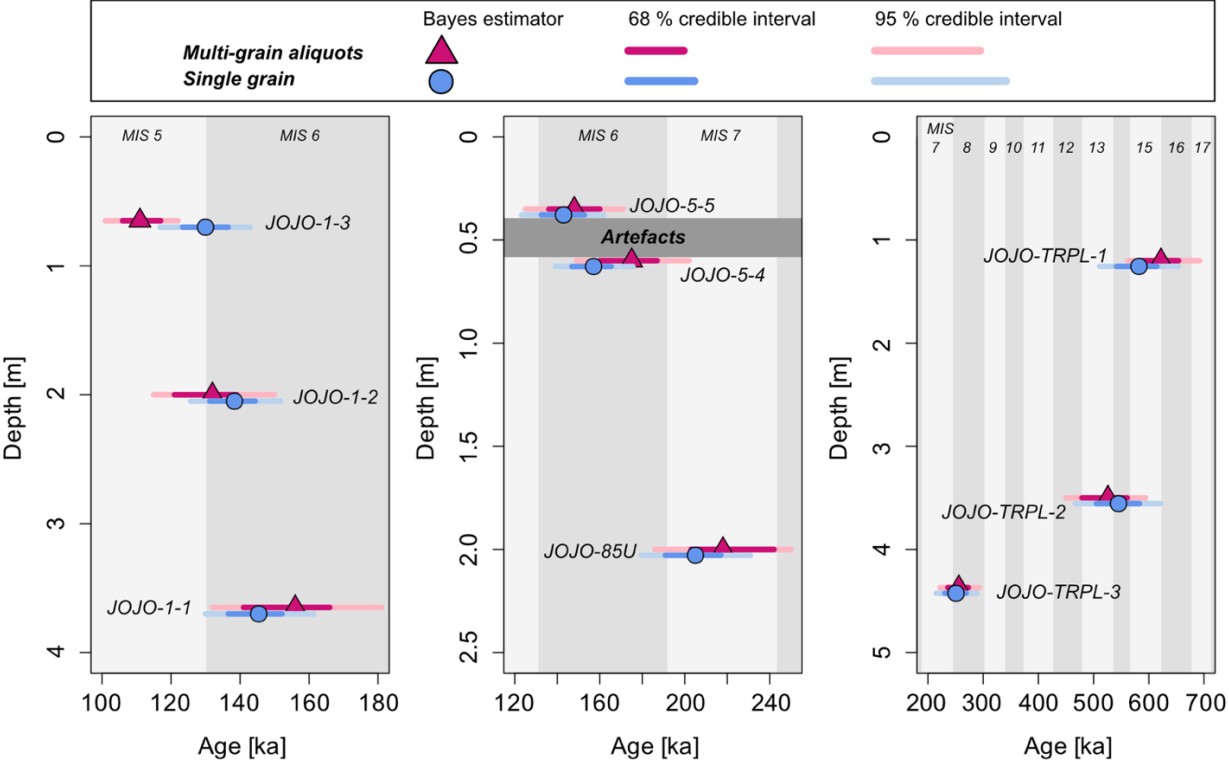

**Fig. 6: Age depth plot of the three sediment profiles measured (profiles Jojosi-1, Jojosi-5, and Jojosi Triple Junction, from left to right). The ages displayed here were obtained using BayLum with stratigraphic constraints. Note: JOJO-TRPL is a cut-and-fill profile, meaning that the depth is not meaningful in terms of stratigraphic order, cf. Fig. 1. The shaded areas indicate marine isotope stages (MIS).**

## 6 Conclusions

This study explored the applicability of different dose models to feldspar single grain, synthetic aliquot and multi-grain aliquot data sets for samples with burial dose ranging from ~80 Gy to ~800 Gy and containing various relative numbers of saturated grains. We applied the Central Age Model (CAM), the



Average Dose Model (ADM), BayLum, and a standardised growth curve (SGC) approach using the interpolation of averaged $L_nT_n$ values onto the SGC to the different data sets and were able to show that,

despite some (systematic) differences, on average, all dose models yield similar results within uncertainties. However, a closer look reveals that the relative number of saturated grains influences the applicability of some dose models. CAM and ADM results of equivalent dose distributions, summarised as frequentists approaches, are unable to include information from saturated grains in their calculation. Excluding these grains biases the data sets. In contrast, BayLum and the CAM SGC $L_nT_n$ approach allow

for the inclusion of saturated grains, with these two methods using different approaches. Despite, on average, good agreement between these two methods, samples with a large number of saturated grains impact the consistency of the result. Overall, all dose models and data sets tested give consistent results below a saturated grain threshold of ~15 %, corresponding to a dose of ~120 Gy (ADM) in this study.

Using the advantage of BayLum to include stratigraphic information in the age calculations, we were able

to establish an internally consistent single grain and multi-grain aliquot-based chronology for the three sampled sections at Jojosi, constraining erosional and depositional processes from ~100 ka to ~700 ka, and placing the human occupation of the area within early MIS 5 and late MIS 6.

**Author Contributions**

**Svenja Riedesel:** Conceptualization, Methodology, Formal analysis, Investigation, Visualization,

Writing – Original Draft. **Guillaume Guérin:** Conceptualization, Methodology, Investigation, Software, Writing - Review & Editing. **Kristina J. Thomsen:** Conceptualization, Investigation, Writing - Review & Editing. **Mariana Sontag-González:** Conceptualization, Software, Writing- Review & Editing. **Matthias Blessing:** Investigation, Writing-Review & Editing. **Greg Botha:** Conceptualization, Writing – Review & Editing. **Max Hellers:** Investigation, Writing - Review & Editing. **Gunther Möller:**

Investigation, Writing-Review & Editing. **Andreas Peffeköver:** Investigation, Writing - Review & Editing. **Christian Sommer:** Conceptualization, Investigation, Writing-Review & Editing. **Anja Zander:** Investigation, Writing - Review & Editing. **Manuel Will:** Conceptualization, Resources, Funding acquisition, Project administration, Writing - Review & Editing



**665   Competing Interests**

The authors declare that they have no conflict of interest.

**Data availability**

The data has been uploaded to Zenodo and is available via doi: 10.5281/zenodo.12759293

**Acknowledgements**

The research at Jojosi is funded by the Deutsche Forschungsgemeinschaft (DFG, ID: 467042592, awarded to MW). SR's research is funded by the European Union's Horizon Europe research and innovation programme (RECREATE, grant no. 101103587). We like to express our gratitude towards Morena Molefe of Batlokoa Ba Molefe and the Tribal Council for granting us permission to do the research in their traditional authority area and for their ongoing support of our work. We would like to thank

Lawrence Msimanga and Hanna Pehnert for their support in the field. Vicki Hansen (Technical University of Denmark) is thanked for her help performing the multi-grain aliquot measurements at Risø. SR would like to thank Kathrin Jung for preparing the epoxy pucks for electron microprobe analyses and Sebastian Kreutzer (Heidelberg University) for rewriting the BayLum code to help with complications which occurred during data analysis.

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
