# Peer review of "A direct comparison of single grain and multi-grain aliquot luminescence dating of feldspars from colluvial deposits in KwaZulu-Natal, South Africa"

_Geochronology, 2024_

## Author Response (AR1)

**Responses to reviewer comments on Preprint gchron-2024-19**

**Reviewer 1 comments on Preprint gchron-2024-19**

*We would like to thank the reviewer for taking the time to carefully review our manuscript and for providing valuable and constructive feedback. We reply to each comment raised individually and address the comments in our revised manuscript.*

Dear editor,

This manuscript presents a systematic comparison of single and multi-grain luminescence dating of feldspars to date colluvial sediments from KwaZulu region in South Africa. The authors comprehensively investigate the saturation of the IR signal on single and multi-grain aliquots. Then, they apply four different dose models to determine the equivalent dose (De) of their samples, including the central age model (CAM), the average dose model (ADM), BayLum, and the standardized growth curve (SGC) approach. The influence of the saturated aliquots on the De is also investigated. At the end they choose the BayLum for age calculation as it includes the saturated grains. The new ages constrain erosional and depositional processes from ~100 ka to ~700 ka, and human occupation in the area to MIS 5-6.

The manuscript is well written and represent an important contribution to the growing knowledge on feldspars behavior and its application in luminescence dating. I recommend receiving this manuscript for publication in Geochronology after addressing the following comments.

*Reply: Thank you very much for your kind and positive feedback.*

General comments

The feldspar samples dated in this study have very low $K_2O$ concentration of <0.5 %. Even 0 for one of the samples. Single grains from two samples were analysed for mineralogy and oxides contents. This presents an opportunity to look into the correlation between the different minerals and signals. The authors looked into the correlation between $K_2O$ and the IR signals (in the supplementary), but not other oxides.

*Reply: Thank you very much for your comment. We chose to focus on the $K_2O$ content for this manuscript, because our main goal is to date the sediments. Since the K-concentration influences the dose rate, it directly affects the ages. We felt that going into further detail on the general geochemistry of the grains would distract from the research focus of the paper.*

The manuscript deals with dating of colluvial sediments. This kind of sediments is hard to bleach. The authors conclude that there is no partial bleaching, based on the appearance of the dose distribution and the bleaching test of Buylaert et al. (2013). I'm not convinced that the samples are bleached. At high doses, the traps are filled in a slower rate, which results in a smoother dose distribution in case of partial bleaching (Fig. 3). In addition, the saturated grains/aliquots, are not represented in the dose distribution. Section 4 describes the amount of the saturated grains in the samples and their influence on the equivalent dose (De). But, the reasons for the presence of the saturated grains are not discussed. If these grains are not bleached, they should not be included in age calculation of the colluvial samples. The final conclusion of preferring BayLum is adequate. Yet, a more in-depth discussion is needed.

*Reply: Thank you for this comment. We agree that displaying the equivalent dose distributions graphically in any form might raise questions, due to the presence of saturated grains, which cannot be visually displayed in these plots. We have thus moved the original figure 3 to the supplementary material. We have further added to the figure caption that these graphs only partially represent the $D_e$ distributions due to saturation issues.*

*To further clarify our interpretation of the bleached nature of the samples, we have adjusted the original figure 4c. Whilst we show all accepted grains from all samples in (a) and (b), we limit the display in (c) to grains with uncertainties on their De < 20 %. We further added uncertainties on the De values in the graphical display. Figure (c) now visually shows these grains and furthermore the same grains, but with their De values corrected for fading. The fading uncorrected ratio of $IRSL_{50}$/post-IR $IRSL_{225}$ De of these grains is 0.85 ± 0.03. If the $IRSL_{50}$ equivalent dose values are fading corrected the ratio rises to 1.16 ± 0.04. This furthermore shows that the grains have been sufficiently reset prior to burial. For further clarification, we now describe this relationship in section 4.1.*

*In section4.4. we already outlined that we tested the $D_0$-based filtering procedure and that excluding finite grains based on the D0 value of their dose response curve does not influence the CAM and ADM results. In the original version we also mentioned that D0 filtering resulted in a decrease of the number of saturated grains in the samples. To make this a little clearer, we modified these sentences (lines 517-523 in the new version) to:*

*"Thus, we tested if excluding grains of certain $D_0$ thresholds affects the CAM and ADM doses for the single grain data set of all samples, but we could not find any effect on the doses calculated. $D_0$ threshold filtering results in the exclusion of some of the saturated grains, indicating that saturation of individual grains results from early saturation of the corresponding dose response curves. Nevertheless, since the $D_0$ filter had no effect on the CAM and ADM doses, we did not apply this additional rejection criterion."*

The geological and archeological context of the dated samples is a bit thin. Please elaborate on the significance of the new ages to the understanding of human presence in the area throughout the Pleistocene. How the ages correspond to the previous works on the local geomorphology.
*Reply: We now added some more archaeological context to the date samples and on the significance of the new ages for future archaeological interpretations. Additional geological contextual description is added to highlight the polyphase erosion/colluvial transport and gully infill evident over a period spanning at least two glacial cycles.*

Specific comments
L32-34: Please rephrase this sentence. Abstract should not contain "suspicions".
*Reply: Thank you. We rephrased the sentence to: "Measurements of feldspar single grains showed low luminescence sensitivity of the individual grains and a variable proportion of grains in saturation."*

L63: Please refer to the location map at the first mention of the study area.
*Reply: We now refer to Figure 1a and b in this sentence.*

L64: Do not site a paper which was not yet accepted by a journal.
*Reply: We are hopeful that the manuscripts which are currently cited, but are still in revision/under review will be published in time. The preprints are also available online and we added the doi to the reference list.*

L63-66: Please add a reference to this sentence.
*Reply: We have slightly adjusted the sentence and have added a reference (Botha, 1996). The synthesis by Botha (1996) described the colluvial stratigraphy, archaeology and provided a geochronological framework for this region.*

L68-69: What is the chronological framework of the colluvial slope processes according to previous works?

*Reply: The references cited in this and the following sentences cover the spectrum of geological and chronologic studies in this region.*

L69: Li and Wintle (1991) investigates the luminescence sensitivity of samples from KwaZulu region. But these samples are not dated in the paper. This reference is not relevant here.
*Reply: We removed the citation.*

L73: Please add a reference at the end of the sentence.
*Reply: We added a reference to Will et al. (2024).*

L113: Please add a rough assessment of the distance from the source rock.
*Reply: The sentence has been modified to include an estimated range of sediment transport distances, considering the short length of the transportational mid slope and the likelihood of polyphase reworking of the sediments through the gully cut-and-fill cycles. It is possible that part of the sediment population has been retained on the slope despite polyphase reworking.*

L114-129: There is a mixture of past, present, and future tenses. Please be consistent.
*Reply: Sentences have been modified to the present tense.*

L127: Please add Roberts and Duller, 2004 for the SGC references.
*Reply: We added the reference here.*

L127-129: Please change to past tense.
*Reply: Done.*

L135: See comment to L64.
*Reply: We cannot see that we have referenced an unpublished paper here. Will et al. 2024 has been published earlier this year.*

L213: You state the signal and background integrals for the single grain measurements, but not for the multi grain. Please add these details.
*Reply: We added the following sentence: "From the obtained single grain luminescence signal the initial 0.2 s were used as signal for further analysis. It was background-corrected using the last 0.4 s."*

L240: Here and in the supplementary it is written that 46 grain were analyzed; in L245 and L247 it is written that 45 grain were measured; in the caption of figure S1a it is written 44 grains. There only 45 grains in tables S1a,b. Please be consistent.
*Reply: Thank you very much for spotting this inconsistency. It should be 45. We corrected this.*

L243-244: Please add a reference to the $K_2O$ contents range (0 wt% to 16.9 wt%).
*Reply: The range is based on weight percent calculation based on stoichiometry. It gives the theoretical minimum and maximum for $K_2O$ in $KAlSi_3O_8$. We have added a statement to the paper.*

L284-285: Please add an example for each type of model.
*Reply: We already provide references for each of the models. We prefer to only cite the original references, which present the models, rather than selecting studies, which made use of the models.*

L350: Please add $IRSL_{50}$ results to the supplementary data (Equivalent to table 4 in the manuscript).
*Reply: The supplementary material now contains a table detailing results for the $IRSL_{50}$ signal, including the relative overdispersion and results obtained using BayLum, to also provide numerical information to the data that was already displayed in Fig. S10.*

L351: According to Table 4, the OD of the SG De distributions for post-IR IRSL$_{225}$ ranges from 26 % (sample JOJO-TRPL-2) to 74 % (sample JOJO-5-5).
*Reply: Thank you very much for spotting this inconsistency. We corrected it. The values in Table 4 are the correct ones.*

L367: According to table 3, the lowest percentage of saturated grains for IRSL$_{50}$ is 0.7 % (sample JOJO-85U).
*Reply: Thank you. Yes, it should read 0.7 %.*

L368-370: Can you add a plot of number of saturated grains vs. SG CAM dose to the supplementary?
*Reply: We have now added a figure displaying this relationship to the supplementary material and we now refer to this figure in the main text, when describing this relationship.*

L380-381: This sentence is not clear.
*Reply: We deleted this sentence and the following sentence.*

L379-382: You start two successive sentences with "However". Please rephrase.
*Reply: We deleted the second sentence, which started with "however".*

L382-385: Although IR$_{50}$ fading is very probable, it does not excludes partial bleaching of the post-IR IRSL$_{225}$ signal, especially if taking into consideration the sedimentary environment.
*Reply: Please see our reply to your general comment. An updated subfigure 3c (was 4c) as well as further details in the text (section 4.1) now hopefully clarify this.*

L398-400: Do you have an explanation to this observation?
*Reply: Unfortunately, we do not have a definitive answer for the different numbers in saturated grains/single aliquots/synthetic aliquots, but we discuss some options later on in the manuscript. Please see section 5.1.*

L410-411: For SG (350-351) you present the OD values range without errors, while for the MG and the SynAl you present the OD values range with errors. Please be consistent.
*Reply: We now added uncertainties to the overdispersion values obtained for the single grain data set.*

L411: According to Table 4, the OD of the SynAl De distributions for post-IR IRSL$_{225}$ ranges from 14 % (sample JOJO-85U) to 40 % (sample JOJO-5-5). Is the confusion in the locations of MG and SynAl in table 4?
*Reply: Thank you for spotting this. We corrected this accordingly.*

L522: Consider to move Fig.S7 to the manuscript, as it has important visualization of the dose response curve characteristics between the different methods.
*Reply: Thank you for this suggestion. We have thought about this already during the preparation of the manuscript, but decided to have the figure in the supplement. Although the figure gives great insights into the saturation behaviour of single grains and multiple grains, we have no definitive answer what causes this and would keep this for further investigation.*

L424-425: Please change to past tense.
*Reply: Corrected.*

L436-438: Please move this information to the table's caption.

*Reply: We have moved the text according to the suggestion.*

L571-574: Please refer to Fig. 5.
*Reply: We now refer to this figure here.*

L604: Generally, in active fluvial systems, sediments older than MIS5 are not common. Please add examples from other places.
*Reply: Examples of dated alluvial deposits from the northern Cape and Free State provinces, described by Claassen (2018) and Tooth et al (2013) contain some MSA archaeological material but only date back to ~MIS 5. We have now added this to the discussion and the additional references to the reference list.*

Figure 1: If the "Luminescence samples not included in this study" do not contribute any relevant information to the manuscript, please remove them from the figure.
*Reply: We removed those samples from the figure.*

Figure 5: Please add the sub figures to the caption (a, b, e, f).
*Reply: Done.*

Figure S4: Are the signals and the dose response curves belong to the same aliquots in each sub figure? If yes, state so. If not, please show corresponding curves.
*Reply: Yes, they are. We mention this now in the figure caption.*

Figure S6: Sub figures e, f are not described in the caption.
*Reply: Thank you for spotting this. We added e and f to the caption.*

Table S1: What is "n" in the table? Is it the number of measurement points per grain? In the caption it is written that "two to six point measured on each grain", but "n" is ranging from 1 to 7. In the caption of Figure S1 it is written "Two to three points were measured per grain".
*Reply: Thank you for raising this. We clarified it in the table caption, where n is now explained and six is exchanged with seven.*

Figure S9: Please add a legend to the figure.
*Reply: We added a description of the different lines to the figure captions.*

Figures S11-S12: Please delete the "<>" in the X-axis title.
*Reply: The axis now only reads "Regenerative dose [Gy]"*

Technical corrections
L70: Teeme et al. (2008) and Lyons et al. (2013) are missing from the references list.
*Reply: Thank you for spotting this. We added these to the reference list.*

L73: Please choose "over" or "for".
*Reply: Thank you. We corrected this.*

L88: Huntley (2006) and Kars et al. (2008) are missing from the references list.
*Reply: Thank you for spotting this. We added these to the reference list.*

L95: Riedesel et al. (2018) is missing from the references list.
*Reply: Added.*

L114: Please change "possibility" to "possibly".
*Reply: Done.*

L123: Please change "following" to "follow".
*Reply: Done.*

L127: You cite Li et al. (2017) in the manuscript, but have Li et al. (2018) in the reference list.
*Reply: We added Li et al. 2017 to the reference list.*

L158: Botter-Jensen et al. (2010) is missing from the references list.
*Reply: We added the reference.*

L246: it is written "max=16.3 wt%", but in table S1b the highest $K_2O$ content is 16.23.
*Reply: Thank you. We corrected it.*

L311: Please change to "There is a possibility".
*Reply: Done.*

L314: Li et al. (2017) is missing from the references list.
*Reply: We added this.*

L316: Please change "an" to "a", and "A" to "The".
*Reply: Done.*

L514: Thomsen et al. (2005) is missing from the references list.
*Reply: Added.*

L567: Please change Fig. "7a" to "S7a".
*Reply: Thank you. We changed it accordingly.*

**Reviewer 2 comments on Preprint gchron-2024-19**

*We would like to thank the reviewer for taking the time to carefully review our manuscript and for providing valuable and constructive feedback. We reply to each comment raised individually and address the comments in our revised manuscript.*

**General comments**

In this study, feldspar single grain and multi grain aliquots were used to date colluvial deposits from KwaZulu-Natal in South Africa using a pIRIR$_{225}$ protocol. A systematic comparison of the De values was conducted using four different age models. The effect of saturated grains on the calculated De values was also investigated for both single grain and multi grain aliquots. Ages derived using BayLum were considered for the final interpretation.

This manuscript is well written and is deserved a publication in Geochronology after addressing the comments.

*Reply: Thank you very much for your kind and positive feedback.*

For the type of environment that this study was conducted in, i.e. colluvium, partial bleaching can be an issue particularly for high temperature IRSL signals as also described in the text. I think that there are indications of partial bleaching present in some of the KDE plots displayed in Fig. 3. The MAM age model can be investigated in such a case and compared with other age models. If saturated grains are present in a sample, how one may know if these are not the partially bleached grains? Also what are the consequences of including them in the final age calculation (if they are actually not well bleached)?

*Reply: We unsuccessfully tested the minimum age model during an earlier phase of our data analysis. We added a paragraph to section 4.4 explaining why we did not consider the MAM further:*

*"We also tested the Minimum Age Model (Galbraith et al., 1999, the logged 3-parameter version) on the single grain data sets. However, the shape of the dose distributions already indicated that the MAM might be inappropriate. Furthermore, tests revealed that p0 values (an indicator of the percentage of grains of the full distribution included in the MAM calculation) were low (< 2 %), indicating the model as unsuitable for our samples. Thus, the MAM results are not discussed further."*

*In addition, we show that incomplete bleaching is unlikely to have occurred, because of the good agreement between IRSL$_{50}$ and post-IRIRSL$_{225}$ doses (please see new Fig. 3c).*

*To address the question regarding the number of saturated grains, we added the following text to the manuscript:*

*Section 4.1:*

*"The two oldest samples, JOJO-TRPL-1 and JOJO-TRPL-2, exhibit the largest relative number of saturated grains for the post-IR IRSL$_{225}$ signal, with 67 % and 48 % of the grains in saturation, respectively. This indicates that both samples are close to the feldspar luminescence dating limit in this area."*

*We have modified figure 3c (which was Fig. 4c) to only display those grains which have D$_e$ uncertainties <20%. We then fading corrected the doses of these grains and display the fading corrected and the un-corrected ratios in this subfigure. Ratios calculated for the uncorrected and corrected doses are 0.85 ± 0.03 and 1.16 ± 0.04, respectively. We also describe this now in section 4.1.*

*We also agree with the reviewer that using KDE plots to visualise the $D_e$ distributions is not ideal. We have thus moved Fig. 3 to the supplementary material and have added a remark in the figure caption that "these figures only partly represent the measurement results due to saturation issues."*

**Specific comments**
Numbers refers to line numbering used in the text.

65: Please add a map or photo.
*Reply: We now refer to Fig. 1c here.*

138, 139: Does this mean some areas were avoided and no samples were collected for dating (e.g. boundaries)
*Reply: Samples were taken with the aim of best capturing the timing of the landscape changes and the archaeology. As stated in the in lines 138-139, dosimetry samples were taken to account for visible changes in the sediment. How these samples were used is described in section 2.3.*

Fig. 1b can be better presented, to give a better view of the area.
*Reply: Fig. 1b is the result of aerial photographs taken during fieldwork. Multiple images were combined to obtain this photo-based map. We cannot modify the images taken during the field campaign.*

188: the error is mentioned to be the standard error, is this the same for the IR50 signal?
*Reply: Yes, it is. We now added "(± standard error)" to the sentence describing the $IRSL_{50}$ fading rates.*

223: Same criteria were mentioned in lines 198-199 for multi grains. So what is considered additional?
*Reply: The criteria are in addition to the ones mentioned above. They were only used for the SGC approach and not for the data extracted using Analyst. We removed "single grain" to make it clear that the emphasis of this sentence is on "SGC".*

244: Please add a reference.
*Reply: The range is based on weight percent calculation based on stoichiometry. It gives the theoretical minimum and maximum for $K_2O$ in $KAlSi_3O_8$. We have added a statement to the paper.*

318: the term "extrapolation" should be avoided here as the whole process is interpolation onto the dose response curve.
*Reply: We have changed the sentence to: "The CAM is then applied to all $L_n/T_n$ values (renormalized according to the SGC) of grains/multi-grain aliquots that passed the acceptance criteria, and the obtained central $L_n/T_n$ (with uncertainties) is projected onto the SGC. The corresponding x-values informs on the dose of the sample."*

338-340: Is there a reason for this difference in the accepted grain for IR50 and pIRIR?
*Reply: Unfortunately, we don't know and we'll leave this up for future investigation.*

Fig. 3d: There is only one accepted synthetic aliquot
*Reply: Yes, indeed. Only one synthetic aliquot gave finite results. Thus, we could not obtain any dose information from the synthetic aliquot data set of sample JOJO-TRPL-1 (cf. Table 4).*

Table 3 caption: Add something like 'as indicated in brackets' after "The relative number of accepted grains"

*Reply: We added the following to the table caption: "The relative n and $n_{sat}$ is given in parenthesis behind the absolute numbers."*

368, 369: This sentence is not clear.
*Reply: We slightly modified the sentence to: "The relative number of saturated grains systematically increases with the size of the single grain CAM dose […]"*

380: It would be good to add the fitted equation.
*Reply: We have modified former Fig. 4 (now Fig. 3) and removed the sentence regarding the fit. The $R^2$ of the linear fit would have been 0.06. However, we think that the modified Figure 3c provides more information on the bleachability of these grains and we explain this further in response to one of your more general comments.*

381, 382: Please rephrase. This sentence is not clear.
*Reply: We deleted this sentence, in accordance with a comment raised by reviewer 1.*

409: Please briefly describe how you calculate this.
*Reply: We did not count the number of grains on the discs. We here refer to an estimate made by Duller (2008). We rephrased the sentence and hope that this clarifies the procedure used.
"[…] whereas for small multi-grain aliquots we summed the signal of approximately 30 grains as suggested by Duller (2008)."*

427: Please avoid the term "non-fading"
*Reply: We changed it to "low-fading exhibiting luminescence signal".*

454: Is this D0 or 2D0?
*Reply: It is $D_0$.*

Fig. 5: There is no description for a, b, e, and f. Please also add some more explanation to this figure caption.
*Reply: Thank you for spotting this. We added the description to these sub-figures.*

645, 646: provide a value instead of using the term "on average"
*Reply: We deleted this sentence altogether.*

647: "give consistent results" please add within what uncertainty.
*Reply: We added "within one standard error" to the sentence.*

**Technical comments**
40: Please check Ln/Tn throughout the text, sometimes it is referred to as LnTn.
*Reply: We use $L_n/T_n$ when we talk about values or ratios, as this is the mathematical expression. However, since Li et al (2020) termed the method $L_nT_n$ method, we decided to stay in line with the original publication whenever we refer to the method applied for dose determination. Nevertheless, there were a few inconsistencies, and we now corrected them.*

68: add a comma after references
*Reply: We added a full stop. Thank you for pointing this out.*

70: use a comma after Clarke et al. (2003)
*Reply: Thank you. We corrected this.*

74, 75: It feels like this sentence is not complete.
*Reply: We corrected the sentence.*

87-89: This sentence should be cut from here, so that "Despite these advantages,…" can be written exactly after the sentence ending in Line 87.
*Reply: We decided to keep the sentence, but modified the following sentence to: "Despite its advantages […]"*

112: Jojosi donga: please check it throughout the text, sometimes it is written with the capital: Donga
*Reply: Thank you. It now reads "Jojosi donga" throughout the manuscript.*

122: c.f.: please check it throughout the text (cf., c.f.)
*Reply: Thank you. Done.*

123: Delete "to" before following
*Reply: It now reads "to follow".*

146: use a comma after "From this fraction"
*Reply: Added.*

165: use a comma after "measurements"
*Reply: No comma is needed here.*

171: use a comma after "tests"
*Reply: No comma is needed here.*

Fig. 2 caption: omit "than" after ">"
*Reply: Done.*

213: omit "around"
*Reply: We changed it to "at".*

241: add 'the' before "luminescent grains"
*Reply: Added.*

241: divided into
*Reply: Changed.*

246: Omit "2.1 ± 0.4 wt%"
*Reply: We deleted the second occurrence of this in the sentence.*

276: add 'dose' before "remaining"
*Reply: Since the explanation give here refers to "remnant doses" in the same sentence, the addition of "dose" is unnecessary here.*

445: Add 'are' after "CAM doses"
*Reply: We modified the sentence to: "These ratios indicate smaller CAM doses compared to ADM doses […]".*

454: less than…
*Reply: We exchanged "less" with "smaller"*

492: add 'and' before "the dose model"
*Reply: Since the last part of the sentence starts with ", as well as…", no "and" is necessary.*

540: Ln/Tn instead
*Reply: Corrected.*

567: Fig. S7a
*Reply: Corrected.*

586: Add 'multi-grain' before "aliquot"
*Reply: Thank you for spotting this. We corrected it.*

605: Replace "by" with 'of': of Clarke et al.
*Reply: Corrected.*

606: the IRSL50 ages in Clarke et al.
*Reply: We added "ages" to the sentence.*

615: Add 'the' before "Middle Pleistocene"
*Reply: We added "the" here.*

623 and 626: Jojosi 1 or Jojosi-1, please check throughout the text
*Reply: It now reads Jojosi 1 throughout the manuscript.*

624: use a comma after "For Jojosi 5"
*Reply: There's no comma needed here.*

639: use a comma after "the SGC"
*Reply: There is no comma needed here.*

642: CAM and ADM equivalent dose distributions results
*Reply: We deleted "results of" in this sentence.*

645: use 'while' instead of "with these two methods"
*Reply: Done.*

**Community Comment on Preprint gchron-2024-19**

*We would like to thank Barbara Mauz for taking the time to carefully review our manuscript and for providing valuable and constructive feedback. We reply to each comment raised individually and address the comments in our revised manuscript.*

This is an empirical study that uses sand-sized K-rich feldspar as dosimeter and compares data obtained from single grain and synthetic aliquots with data obtained from multi-grain aliquots using the $pIRIR_{225}$ measurement protocol. Then the data are quantified using four different dose distribution models. This sounds like an approach with (too?) many parameters and, often, such papers are hard to read. Not this paper, I feel, because It is well-structured and methods employed are well described. I was particularly interested to learn how this study evaluates samples close to saturation level but this turned out to be a bit difficult, hence my first question (see below). While reading in order to find out how saturation was defined, a couple of other questions popped up, which are summarised below.
*Reply: Thank you very much for this feedback and your general positive opinion on our manuscript.*

Saturation dose and D0 – section 2 describes a lot of details (some multiple times, e.g. filter) but not how these two values were determined. I gather from the text that a grain/aliquot exhibit saturation when Ln/Tn > $I_{max}$ where $I_{max}$ is the "maximum asymptote of the dose response curve" (line 312). The plots suggest that $beta_{max}$ = 800 Gy, but the D0 plot (Fig. S7a) shows D0 >500 Gy for a significant number of grains/aliquots. This suggests that maximum beta dose does not correspond to $I_{max}$. I am therefore wondering if the difference between D0-single grain and D0-multi grain aliquot is caused by the difference between $I_{max}$ and $beta_{max}$?
*Reply: The doses used for the dose response curves where the same for single grain and multi-grain measurements, except for the two oldest samples, where further dose points were needed to describe the dose response curve of the multi-grain dataset. In addition, we now explicitly state how $D_0$ values are determined (l. 195-196): '(L/T = Imax. (1 – exp (-D/D$_0$), where L/T is the normalised OSL signal, D is the laboratory dose and $D_0$ a curvature parameter)' for multi-grain aliquots. We also refer to this equation for single-grain aliquots (l. 217-218: 'A single saturating exponential function (similar to that used for multi-grain aliquots, see previous paragraph) was used to fit L$_x$/T$_x$ values obtained for single grains. '– our addition in brackets. Finally, we also define 'saturated aliquots' based on this function (note: this definition already was in our initial text, but we moved it next to the definition of dose response curves, l. 218-222): 'Using such curve fitting functions, following earlier studies (e.g., Heydari and Guérin, 2018; Chapot et al., 2022; Arce-Chamorro and Guérin, 2024) we regard grains as saturated when the L$_n$/T$_n$ ratio and/or the sum of this ratio plus its uncertainty does not intercept the dose response curve, thus lies above Imax. All grains which yield a finite dose ± finite uncertainties are regarded as not saturated in this study.' We now hope that our text is clearer; we also would like to state that since the function used for curve fitting is asymptotic, strictly speaking there is no dose for which I = Imax: It only tends towards Imax.*

BayLum – it includes aliquot data exhibiting Ln/Tn > $I_{max}$ and fits the single exponential function to the Lx/Tx data. How was $D_e$ determined if interpolation is not an option? A Gauss distribution of individual doses around De was assumed also for those samples with a high $D_e$ (e.g. >150 Gy)?
*Reply: This question is – rightfully – often raised. For a detailed explanation, we refer the reader to the supplementary material of Arce-Chamorro and Guérin (2024), appendix A7. To summarise: in BayLum, individual $D_e$ estimates are not parameterised. While in classical analysis, a pair of (x,y) values define the equivalent dose of an aliquot by x ± y (corresponding to a Gaussian probability density function), in BayLum probability density functions can take any form. As a result, a saturated aliquot considered alone will, in BayLum, yield an approximately constant probability density (i.e., any 'high-dose' value – the notion of high dose depending on D0 – will be equally likely). When this*

*aliquot is taken within a sample, the likely high dose values will be constrained by the other aliquots, because all De values are assumed to form a Gaussian distribution.*

Grain size of the dosimeter – you selected 200-250 micron grain size - did you assume that this coarse fraction is better bleached than, say 90-150 micron grains, because grains did roll downslope individually and were not part of the bed load? The photos in Fig. 1 suggest transport of sediment by gravity and/or by water, each mode generating a different degree of bleaching across grain size fractions.
*Reply: The grain size was chosen as the coarser fraction was expected to be better bleached and for practical reasons, as it is easier to mount these grains in the single grain discs and to ensure that only a single grain is in each hole of the disc.*

The CLL calibrates the beta source every 6 month, presumably using the same calibration quartz sample each time, i.e. quartz grains that have received the same gamma dose (e.g. 5 Gy). How do you fit a regression line through datapoints that have approximately the same value? Is the intercept always zero?
*Reply: A decay is visible, because most of the calibrations recorded and used in the spreadsheet (at least for the older readers) go back to the early 2010s. Here the decay of the $^{90}Sr/^{90}Y$ source is clearly visible. Furthermore, in those cases, variations and outliers are readily spottable and here the fit provides a more robust estimate compared to using the result of a single calibration.*

Introduction - you say that a number of studies exists already on colluvial deposits in S-Africa – I am wondering which open questions did these studies leave behind and how did these papers guide your study?
*Reply: The interest of overall study this manuscript belongs to is of mostly archaeological interest. The project tries to understand landscape use, behavioural adaptions and the cultural evolution of the MSA hunter-gatherers in eastern South Africa. As we mention in the introduction, other donga sites have been investigated and were also subject to luminescence dating studies. Here we not only use the luminescence chronology to infer past landscape changes, but also to establish a chronology of the archaeological findings at these open-air sites. This manuscript presents the technical details of the luminescence dating approach used at Jojosi, and in the further course of the project more information on the archaeological and palaeo-geographical findings, supported by further luminescence dating, will be made accessible. We have now added some further details to the introduction and the discussion (especially section 5.2) where we outline the implications of the here presented work.*

Gamma spectrometry - this may sound like my personal hobbyhorse, but since Murray et al. (2015; interlab comparison study) we know that we have an issue with gamma spectrometry. As a consequence, I think we should report details (measurement time and geometry, calibration, peak selection) in publications.
*Reply: The CLL successfully participated in the laboratory intercomparison, and we now added further details to the gamma spectrometry performed at the CLL to the Materials and Methods section of the paper. The section now reads:*

*To estimate the external dose rate, uranium (U), thorium (Th) and potassium (K) contents were determined by high-resolution gamma spectrometry. Approximately 200 g of dried, homogenised sediment was stored in an airtight box, filled to max. capacity, for at least four weeks to compensate for radon loss induced by sample preparation, before measurement with an Ortec Profile MSeries GEM Coaxial P-type high-precision Germanium Gamma-Ray detector. The gamma spectrometers at the CLL are calibrated at least every three months. $^{60}Co$ and $^{152}Eu$ standards are used for the energy calibration and a Nussi sediment standard (Preusser and Kasper, 2001) is used for efficiency*

*calibrations, with the measurement results being compared to the updated concentrations determined by Murray et al. (2018). The gamma samples in this study were measured for 200,000 s. Peak selection for activity calculations included the following peaks: $^{232}$Th decay series: 338 keV, 911 keV, 969 keV, 239 keV, and 583 keV; $^{238}$U decay series: 295 keV; 352 keV; 609 keV, 1765 keV; $^{40}$K: 1461 keV.*

The term "palaeodose" means past radiation dose. I suggest to follow Huntley (2001; Ancient TL): "in our work it is not the actual past radiation dose that is determined, but the beta or gamma dose that results in the same luminescence intensity during thermal or optical excitation" where "the same" is the fundamental assumption.

*Reply: Thank you for this comment. We changed the instances where we used the term "palaeodose".*